# Satellite-based Estimation of the Impacts of Summertime Wildfires on
# PM$_{2.5}$ concentration in United States
## Zhixin Xue[1], Pawan Gupta[2,3], and Sundar Christopher[1]
[1]Department of Atmospheric and Earth Science, The University of Alabama in Huntsville,
Huntsville, 35806 AL, USA
[2]STI, Universities Space Research Association (USRA), Huntsville, 35806 AL, USA
[3]NASA Marshall Space Flight Center, Huntsville, AL, 35806, USA
**Abstract**
Frequent and widespread wildfires in North Western United States and Canada has become
the "*new normal*" during the northern hemisphere summer months, which significantly degrades
particulate matter air quality in the United States. Using the mid-visible Multi Angle
Implementation of Atmospheric Correction (MAIAC) satellite-derived Aerosol Optical Depth
(AOD) with meteorological information from the European Centre for Medium-Range Weather
Forecasts (ECMWF) and other ancillary data, we quantify the impact of these fires on fine
particulate matter concentration (PM2.5) air quality in the United States. We use a Geographically
Weighted Regression method to estimate surface PM2.5 in the United States between low (2011)
and high (2018) fire activity years. Ourresults achieve an overall Leave One Out Cross Validation
(LOOCV) R$^2$ value of 0.797 with RMSE between 3~5 $\mu g\ m^{-3}$. Our results indicate that smoke
aerosols caused significant pollution changes over half of the United States. We estimate that
nearly 29 states have increased PM2.5 during the fire active year and 15 of these states have PM2.5
concentrations more than 2 times than that of the inactive year. Furthermore, these fires increased
the daily mean surface PM2.5 concentrations in Washington and Oregon by 38 to 259μgm$^{-3}$ posing
significant health risks especially to vulnerable populations. Our results also show that the GWR
model can be successfully applied to PM2.5 estimations from wildfires thereby providing useful
information for various applications including public health assessment.

## 1. Introduction

The United States (US) Clean Air Act (CAA) was passed in 1970 to reduce pollution levels
and protect public health that has led to significant improvements in air quality (Hubbell et al.,
2010; Samet, 2011). However, the northern part of the US continues to experience an increase in
surface PM2.5 due to fires in North Western United States and Canada (hereafter NWUSC)
especially during the summer months and these aerosols are a new source of 'pollution' (Coogan
et al., 2019; Dreessen et al., 2016).  The smoke aerosols from these fires increase fine particulate
matter (PM2.5) concentrations and degrade air quality in the United States (Miller et al., 2011).
Moreover several studies have shown that from 2013 to 2016, over 76% of Canadians and 69% of
Americans were at least minimally affected by wildfire smoke (Munoz-Alpizar et al., 2017).
Although wildfire pre-suppression and suppression costs have increased, the number of large fires
and the burnt areas in many parts of western Canada and the United States have also increased.
(Hanes et al., 2019; Tymstra et al., 2019). Furthermore, in a changing climate, as surface
temperature increases and humidity decreases, the flammability of land cover also increases, and
thus accelerate the spread of  wildfires (Melillo et al., 2014). The accumulation of flammable
materials like leaf litter can potentially trigger severe wildfire events even in those forests that
hardly experience wildfires (Calkin et al., 2015; Hessburg et al., 2015; Stephens, 2005).
Wildfire smoke exposure can cause small particles to be lodged in lungs that may lead to
exacerbations of asthma chronic obstructive pulmonary disease (COPD), bronchitis, heart disease
and pneumonia (Apte et al., 2018; Cascio, 2018). According to a recent study, a 10 $\mu g m^{-3}$
increase in PM2.5 is associated with a 12.4% increase in cardiovascular mortality (Kollanus et al.,
2016). In addition, exposure to wildfire smoke is also related to massive economic costs due to
premature mortality, loss of workforce productivity, impacts on the quality of life and
compromised water quality (Meixner and Wohlgemuth, 2004).
Surface $PM_{2.5}$ is one of the most commonly used parameters to assess the health effects of
ambient air pollution. Given the sparsity of measurements in many parts of the world, it is not
possible to use interpolation techniques between monitors to provide $PM_{2.5}$ estimates on a square
kilometer basis. Since surface monitors are limited, satellite data has been used with numerous
ancillary data sets to estimate surface $PM_{2.5}$ at various spatial scales. Several techniques have been
developed to estimate surface $PM_{2.5}$ using satellite observations from regional to global scales
including simple linear regression, multiple linear regression, mixed-effect model, chemical
transport model (scaling methods), geographically weighted regression (GWR), and machine
learning methods (see Hoff and Christopher, 2009 for a review). The commonly used global
satellite data product is the 550nm (mid-visible) aerosol optical depth (AOD) which is a unitless
columnar measure of aerosol extinction. Simple linear regression method uses satellite AOD as
the only independent variable, which shows limited predictability compared to other methods and
correlation coefficients vary from 0.2 to 0.6 from the Western to Eastern United States (Zhang et
al., 2009).  Multiple linear regression method uses meteorological variables along with AOD data,
and the prediction accuracy varies with different conditions including the height of boundary layer
and other meteorological conditions (Goldberg et al., 2019; Gupta and Christopher, 2009b; Liu et
al., 2005). For both univariate model and multi-variate models, AOD shows stronger correlation
with $PM_{2.5}$ during-fire episodes compared to pre-fire and post-fire periods (Mirzaei et al., 2018).
Chemistry transport models (CTM) that scale the satellite AOD by the ratio of $PM_{2.5}$ to AOD
simulated by models can provide $PM_{2.5}$ estimations without ground measurements, which are
different than other statistical methods (Donkelaar et al., 2019, 2006). However, the CTM models
that depend on reliable emission data usually show limited predictability at shorter time scales,
and is largely useful for studies that require annual averages (Hystad et al., 2012). Different
machine learning methods including neuron network, random forest, and deep belief networks
show  improvements on prediction accuracy (with CV $R^2$ values larger than 0.8) which is hard to
accomplish for other parametric regression models (Hu et al., 2017; Li et al., 2017; Wei et al.,
2021, 2020, 2019). However, these methods also require large amount of samples to train the
model which means it is more suitable for daily PM2.5 estimation rather than short-term wildfire
events with relative low occurrence frequency.

The relationship among PM$_{2.5}$, AOD and other meteorological variables is not spatially

consistent (Hoff and Christopher, 2009; Hu, 2009). Therefore, methods that consider spatial
variability can replicate surface PM$_{2.5}$ with higher accuracy. One such method is the GWR, which
is a non-stationary technique that models spatially varying relationships by assuming that the
coefficients in the model are functions of locations (Brunsdon et al., 1996; Fotheringham et al.,
1998, 2003). In 2009, satellite-retrieved AOD was introduced in the GWR method to predict
surface PM$_{2.5}$ (Hu, 2009) followed by the use of meteorological parameters and land use
information (Hu et al., 2013). Meteorological variables are crucial for simulating surface PM$_{2.5}$
since they interact with PM$_{2.5}$ through different processes which will be discussed in detail in the
data section (Chen et al., 2020). Several studies  (Guo et al., 2021; Ma et al., 2014; You et al.,
2016a) successfully applied the GWR model in estimating PM$_{2.5}$ in China by using AOD and
meteorological features as predictors. Similar to all the statistical methods, however, the GWR
relies on adequate number and density of surface measurements (Chu et al., 2016; Gu, 2019; Guo
et al., 2021), underscoring the importance of adequate ground monitoring of surface PM$_{2.5}$.
In this paper, we use satellite data from the Moderate Resolution Imaging
Spectroradiometer (MODIS) and surface $PM_{2.5}$ data combined with meteorological and other
ancillary information to develop and use the GWR method to estimate $PM_{2.5}$. The use of the GWR
method is not novel and we merely use a proven method to estimate surface $PM_{2.5}$ from forest
fires. We calculate the change in $PM_{2.5}$ between a high fire activity (2018) with low fire activity
(2011) periods during summer to assess the role of NWUSC wildfires on surface $PM_{2.5}$ in the
United States. The paper is organized as follows: We describe the data sets used in this study
followed by the GWR method. We then describe the results and discussion followed by a summary
with conclusions.

**2. Data**
A 17-day period (August $9^{th}$ to August $25^{th}$) in 2018 (high fire activity) and 2011 (low fire
activity) was selected based on analysis of total fires (details in methodology section) to assess
surface $PM_{2.5}$ (Table 1).
**2.1 Ground level $PM_{2.5}$ observations:** Daily surface $PM_{2.5}$ from the Environment Protection
Agency (EPA) are used in this study. These data are from Federal Reference Methods (FRM),
Federal Equivalent Methods (FEM), or other methods that are to be used in the National Ambient
Air Quality Standards (NAAQS) decisions. A total of 1003 monitoring sites in the US are included
in our study with 949 having valid observations in the study period in 2018, and a total of 873 sites
with 820 having valid observations in the study period in 2011. $PM_{2.5}$ values less than 2 $\mu gm^{-3}$ are
discarded since they are lower than the established LDL-Lower Detection Limit  (EPA, 2018,

2011).

**2.2 Satellite Data:** AOD which represents the total column aerosol mass loading is related to
surface PM$_{2.5}$ as a function of aerosol vertical properties and physical properties (Koelemeijer et
al., 2006):
$$AOD = PM_{2.5}\, H\, f(RH) \frac{3 Q_{ext,dry}}{4 \rho\, r_{eff}} = PM_{2.5}\, H\, S \qquad (1)$$

Where H is the aerosol layer height, f(RH) is the ratio of ambient and dry extinction
coefficients, Q$_{ext,dry}$ is the extinction efficiency under dry conditions, r$_{eff}$ is the particle effective
radius, $\rho$ is the aerosol mass density and S is the specific extinction efficiency (m$^2$ g$^{-1}$) of the
aerosol at ambient conditions. Therefore AOD usually has a strong positive correlation with PM$_{2.5}$,
and the relationship varies depending on other meteorological parameters which will be discussed
in detail in the following section.
The MODIS mid visible AOD from the Multi-Angle Implementation of Atmospheric
Correction (MAIAC) product (MCD19A2 Version 6 data product) is used in this study. We used
the MAIAC- retrieved Terra and Aqua MODIS AOD product at 1 km pixel resolution (Lyapustin
et al., 2018).  Different orbits are averaged to obtain mean daily values. Since thick smoke plumes
generated by wildfires can be misclassified as cloud, we preserve possible cloud contaminated
pixels to preserve the thick smoke pixels, and only AOD less than 0 will be discarded. Validation
with AERONET studies show that 66% of the MAIAC AOD data agree within $\pm 0.5 \sim \pm 0.1$ AOD
(Lyapustin et al., 2018). Largely due to cloud cover, grid cells may have limited number of AOD
observations within a certain period. On average, cloud free AOD data are available about 40% of
the time during August 9$^{th}$ to August 25$^{th}$ in 2018 when fires were active in the region bounded by
25~50°N, 65~125°W. Smoke flag from the same product is used as a predictor in estimating
surface PM$_{2.5}$. The smoke detection is performed using MODIS red, blue and deep blue bands, and
smoke pixels are separated from dust and clouds based on absorption parameter, size parameter
and thermal thresholds (see Lyapustin et al., 2012; 2018 for further discussion). Smoke flag data
can provide the percentage of smoke pixel in each grid, which is related to smoke coverage.
We also use the MODIS level-3 daily FRP (MCD14ML, fire radiative power) product
which combines Terra and Aqua fire products to assess wildfire activity. The fire radiative energy
indicates the rate of combustion and thus FRP can be used for characterizing active fires (Freeborn
et al, 2014). For purposes of the study we sum the FRP within every 2.3°×3.5° box to represent
the total fire activity in different locations.
**2.3 Meteorological data:** Meteorological information including boundary layer height (BLH), 2m
temperature (T2M), 10m wind speed (WS), surface relative humidity (RH) and surface pressure
(SP) are obtained from the European Centre for Medium-Range Weather Forecasts (ECMWF)
reanalysis (ERA5) product, with a spatial resolution of 0.25 degrees and temporal resolution of 1
hour and is matched temporally with the satellite overpass time. The meteorological parameters
provide important information of different processes affecting surface $PM_{2.5}$ concentration, which
can also be seen as supplements of the AOD-$PM_{2.5}$ relationship as previously discussed.
The BLH can provide information of aerosol layer height (H in equation 1) as aerosols are often
found to be well-mixed within the boundary layer (Gupta and Christopher, 2009b). With same
amount of pollution within the boundary layer, the higher the BLH is, the more $PM_{2.5}$ is distributed
within that layer and vice-versa (Miao et al., 2018; Zheng et al., 2017). Therefore, $PM_{2.5}$ usually
has an anticorrelation with BLH. However, for wildfire events, the aerosol layer height is
sometimes higher than the BLH (Haarig et al., 2018), which leads to lower correlation between
AOD and $PM_{2.5}$ since we use only BLH to present the aerosol layer height. Thus BLH can provide
aerosol vertical information in most cases except for suspended high-layer aerosol caused by fires,
which leads to higher bias of the model for high-layer aerosols near the fire sources. Surface
temperature (T2M) can affect $PM_{2.5}$ through convection, evaporation, temperature inversion and
secondary pollutants generation processes (Chen et al., 2020). The first two processes are
negatively related to $PM_{2.5}$ concentration: 1) higher temperature increases turbulence and
atmospheric convections which accelerate the pollution dispersion ($PM_{2.5}$ decreases); 2) higher
temperature increases evaporation loss of $PM_{2.5}$ including ammonium nitrate and other volatile or
semi-volile components (Wang et al., 2017). The later two processes are positively related to $PM_{2.5}$
by limiting vertical motion and promoting photochemical reactions under high temperature (Xu et
al., 2019; Zhang et al., 2015). Wind speed (WS) are often negatively related to $PM_{2.5}$ since it
increases the dispersion of pollutants. However, unique geographical conditions (such like
mountains) with certain wind directions can cause accumulations of pollutants (Chen et al., 2017).
RH may promote hygroscopic growth of particles to increase $PM_{2.5}$ (Trueblood et al., 2018; Zheng
et al., 2017), but it can also reduce $PM_{2.5}$ through the deposition process. SP may influence the
diffusion or accumulation of pollutants through formation of low-level wind convergence (You et
al., 2017). Precipitation is another factor that largely influences surface $PM_{2.5}$ since it can
accelerate the wash-out of suspended particles, but AOD values are not available when clouds are
present.

**3. Methodology**

To assess the impact of NWUSC fires on $PM_{2.5}$ in the United States, we first estimate the
$PM_{2.5}$ over the study region during a time period with high fire activity (2018). We then use the
same method during a year with low fire activity (2011) to compare the differences between the
two years. The two years are selected based on the total FRP in August calculated within Canada
(49~60°N, 55~135°W) and Northwestern (NW) US (35~49°N, 105~125°W). Table 2 shows the
total FRP in Canada and Northwestern US in August from 2010 to 2018. The total FRP in the two
regions is lowest in 2011 and highest in 2018 during the 9 years, which provides the basis for the
study. In order to create a 0.1° surface $PM_{2.5}$, the GWR model is used to estimate the relationships
of $PM_{2.5}$ and AOD. Detailed processing steps for GWR model are shown in Figure 1.
**3.1 Data preprocessing:** The first step is to resample all datasets to a uniform spatial resolution
by creating a 0.1° resolution grid covering the Continental United States. During this process, we
collocate the $PM_{2.5}$ data and average the values if there is more than one value in one grid. Then
the MAIAC AOD and smoke flagare averaged into 0.1° grid cells. Meteorological datasets are
also resampled to the 0.1° grid cells by applying the inverse distance method.
**3.2 Time selecting & averaging:** Next we select data where AOD and ground $PM_{2.5}$ are both
available (AOD > 0 and $PM_{2.5}$ > 2.0 $\mu g\ m^{-3}$) and average them for the study period (since LDL
of for the FRM method is 2 $\mu g\ m^{-3}$ in 2011 and 3 $\mu g\ m^{-3}$ in 2018, we decides to use the LDL for
2011) (EPA, 2018, 2011). This is to ensure that the AOD, $PM_{2.5}$ and other variables match with
each other, because $PM_{2.5}$ is not a continuous measurement for some sites and AOD have missing
values due to cloud cover and other reasons. Therefore, it is important to use data from days where
both measurements are available to avoid sampling biases.
**3.3 GWR model development and validation:** The Adaptive bandwidth selected by the Akaike's
Information Criterion (AIC) is used for the GWR model (Loader, 1999). For locations that already
have $PM_{2.5}$ monitors, we calculate the mean AOD of a 0.5×0.5° box centered at the ground location
and estimate the GWR coefficients (β) for AOD and meteorological variables to estimate $PM_{2.5}$.
The model structure can be expressed as:
$$PM_{2.5i} = \beta_{0,i} + \beta_{1,i}AOD_i + \beta_{2,i}BLH_i + \beta_{3,i}T2M_i + \beta_{4,i}U10M_i + \beta_{5,i}RH_{sfci} + \beta_{6,i}SP_i + \beta_{7,i}SF_i$$
$$+ \varepsilon_i$$
where $PM_{2.5i}$ $(\mu g\ m^{-3})$ is the selected ground-level PM2.5 concentration at location $i$; $\beta_{0,i}$
is the intercept at location $i$; $\beta_{1,i} \sim \beta_{8,i}$ are the location-specific coefficients; $AOD_i$ is the resampled
AOD selected from MAIAC daily AOD data at location $i$; $BLH_i, T2M_i, U10M_i, RH_{sfci}, SP_i$ are
selected meteorological parameters (BLH, T2M, WS, RH and PS) at location $i$; $SF_i$ (%) is the
resampled smoke flag data at location i and $\varepsilon_i$ is the error term at location $i$.
We perform the Leave One Out Cross Validation (LOOCV) to test the model predictive
performance (Kearns and Ron, 1999). Since the GWR model relies on adequate number of
observations, the prediction accuracy will be lower if we preserve too much data for validation.
Therefore, we choose the LOOCV method, which preserve only one data for validation at a time
and repeat the process until all the data are used. In addition, $R^2$ and RMSE are calculated for both
model fitting and model validation process to detect overfitting. Model overfitting will lead to low
predictability, which means it fits too close to the limited number of data to predict for other places
and will cause large bias.
**3.4 Model prediction:** While predicting the ground-level PM2.5 for unsampled locations, we make
use of the estimated parameters for sites within a 5° radius to generate new slopes for independent
variables based on the spatial weighting matrix (Brunsdon et al., 1996). The closer to the predicted
location, the closer to 1 the weighting factor will be, while the weighting factor for sites further
than the 5° in distance is zero. It is important to note that AOD and other independent variables
used for prediction in this step are averaged values for days that have valid AOD, which is different
from the data used in the fitting process since PM2.5 is not measured every day in all locations.
**4. Results and Discussion**
We first discuss the surface PM$_{2.5}$ for a few select locations that are impacted by fires
followed by the spatial distribution of MODIS AOD and the FRP for August 2018. We then assess
the spatial distribution of surface PM$_{2.5}$ from the GWR method. The validation of the GWR method
is then discussed. To further demonstrate the impact of the NWUSC fires on PM$_{2.5}$ air quality in
the United States, we show the spatial distribution of the difference between August 2018 and
August 2011. We further quantify these results for ten US EPA regions.
**4.1 Descriptive statistics of satellite data and ground measurements**
The 2018 summertime Canadian wildfires started around the end of July in British
Columbia and continued until mid-September. The fires spread rapidly to the south of Canada
during August, causing high concentrations of smoke aerosols to drift down to the US and affecting
particulate matter air quality significantly. From late July to mid-September, wildfires in the
northwest US that burnt forest and grassland also affected air quality. Starting with the Cougar
Creek Fire, then Crescent Mountain and Gilbert Fires, different wildfires in in NWUSC caused
severe air pollution in various US cities. Figure 2a shows the rapid increase in PM$_{2.5}$ of selected
US cities from July 1$^{st}$ to August 31$^{st}$, due to the transport of smoke from these wildfires. For all
sites, July had low PM$_{2.5}$ concentrations (<10 $\mu g\ m^{-3}$) and rapidly increases as fire activity
increases. Calculating only from the EPA ground observations, the mean PM$_{2.5}$ of the 17 days for
the whole US is 13.7 $\mu g\ m^{-3}$ and the mean PM$_{2.5}$ for Washington (WA) is 40.6$\mu gm^{-3}$, which
indicates that the PM pollution is concentrated in the northwestern US for these days. This trend
is obvious when comparing the mean PM$_{2.5}$ of all US stations (black line with no markers) and the
mean PM$_{2.5}$ of all WA stations (grey line with no markers). Ground-level PM$_{2.5}$ reaches its peak
between August 17$^{th}$-21$^{st}$ and daily PM$_{2.5}$ values during this time period far exceeds the 17-day
mean PM$_{2.5}$. For example, mean PM$_{2.5}$ in WA on August 20th is 86.75 $\mu g\ m^{-3}$, which is more
than two times the 17-day average of this region. On August 19[th], Omak which is located in the
foothills of the Okanogan Highlands in WA had PM$_{2.5}$ values exceed 250 $\mu g\ m^{-3}$. According to
a review of US wildfire caused PM$_{2.5}$ exposures, 24-h mean PM$_{2.5}$ concentrations from wildfires
ranged from 8.7 to 121 $\mu g\ m^{-3}$, with a 24 h maximum concentration of 1659 $\mu g\ m^{-3}$ (Navarro et
al., 2018).
Table 3 shows relevant statistics of 15 states that have at least one daily record of non-
attainment of EPA standard (>35 $\mu g\ m^{-3}$ . From the frequency records of non attainment in the
17-day period (last column), four states (Montana, Washington, California and Idaho) were
consistently affected by the wildfires, and large portion of ground stations in these states were
influenced by smoke aerosols. Most of the neighboring states also suffered from short-term but
broad air pollution (third column). Noticeable from these records is that the total number of ground
stations in some of the highly affected states (such as Idaho) is not sufficient for capturing the
smoke. Although there are total 8 EPA stations in Idaho, only two of them have consistent
observations during the fire event; the other two stations have no valid observations, and the
remaining four stations have only 2~6 observations during the 17-day period. Limited valid data
along with unevenly distributed stations makes it hard to quantify smoke pollution in Northwestern
US during the fire event period. Therefore, we utilize satellite data to enlarge the spatial coverage
and estimate pollution at a finer spatial resolution.
The spatial distribution of AOD shown in Figure 2b indicates that the smoke from Canada
is concentrated mostly in Northern US states such as WA, Oregon, Idaho, Montana, North Dakota
and Minnesota. The black arrow shows the mean 800hPa-level mean wind for 17 days, and the
length of the arrow represents the wind speed in ms$^{-1}$. Also shown in Figure 2b are wind speeds
close to the fire sources which are about 4~5 ms$^{-1}$, and according to the distances and wind
directions, it can take approximately 28~36 hours for the smoke to transport southeastward to
Washington state. Then the smoke continues to move east to other northern states such as Montana
and North Dakota. In addition, the grey circle represents the total fire radiative power (FRP) of
every 2.3×3.5-degree box. The reason for not choosing a smaller grid for the FRP is to not clutter
Figure 2b with information from small fires. The bigger the circle is, the stronger the fire is in that
grid and different sizes and its corresponding FRP values are shown in the lower right corner. It is
clear that the strongest fires in 2018 are located in the Tweedsmuir Provincial Park of British
Columbia in Canada (53.333N, 126.417W). The four separate lightning-caused wildfires burnt
nearly 301,549 hectares of the boreal forest. The total FRP of August 2018 in Canada is about
5362 (*1000 MW), while the total FRP of August 2011 in Canada is 48 (* 1000 MW). The 2011
fire was relatively weak compared to the 2018 Tweedsmuir Complex fire and we therefore use the
2011 air quality data as a baseline to quantify the 2018 fire influence on PM$_{2.5}$ in the United States.

**4.2 Model Fitting and validation**

The main goal for using GWR model is to help predict the spatial distribution of PM$_{2.5}$ for

places with no ground monitors while leveraging the satellite AOD and therefore it is important to
ensure that the model is robust. Figure 3a and 3b show the results for 2018 for GWR model fitting
for the entire US and the LOOCV models respectively. The color of the scatter plots represents
the probability density function (PDF) which calculates the relative likelihood that the observed
ground-level PM$_{2.5}$ would equal the predicted value. The lighter the color is, the more points are
present, with a higher correlation. The model fitting process estimates the slope for each variable
and therefore the model can be fitted close to the observed PM$_{2.5}$ and using this estimated
relationship we are able to assess surface PM$_{2.5}$ using other parameters at locations where PM$_{2.5}$
monitors are not available. The LOOCV process tests the model performance in predicting PM2.5.
If the results of LOOCV has a large bias from the model fitting, then the predictability of the model
is low. Higher $R^2$ difference and RMSE difference value indicate that the model is overfitting and
not suitable. The $R^2$ for the model fitting is 0.834, and the $R^2$ for the LOOCV is 0. 797; the RMSE
for the GWR model fitting is 3.46 $\mu g\ m^{-3}$, and for LOOCV the RMSE is 3.84 $\mu g\ m^{-3}$. There are
minor differences between fitting $R^2$ and validation $R^2$ (0.037) and between fitting RMSE and
validation RMSE (0.376 $\mu g\ m^{-3}$) suggesting that the model is not over-fitting and has stable
predictability further indicating that the model can predict surface PM2.5 reliably. In addition, we
also performed a 20-fold cross validation by splitting the dataset into 20 consecutive folds, and
each fold is used for validation while the 19 remaining folds form the training set. The 20-fold
cross validation has $R^2$ of 0.745 and RMSE of 4.3 $\mu g\ m^{-3}$. The increase/decrease in the cross
validated $R^2$ and RMSE indicates the importance of sufficient data used for fitting since a small
decrease in the number of fitting data can reduce the model prediction accuracy. Overall, the
prediction error of the model is between 3~5 $\mu g\ m^{-3}$, which is a reasonable error range for 17-day
average prediction of PM2.5. For data greater than the EPA standard (35 $\mu g\ m^{-3}$), the model has
a RMSE of 12.07 $\mu g\ m^{-3}$, which is a lot larger than the RMSE when using the entire model.
Therefore, the model has a tendency for underestimating PM2.5 exceedances by around 12.07
$\mu g\ m^{-3}$. The larger the PM2.5 is, the greater the model underestimates. To examine the model
performance for high and low polluted areas, the results are divided into two parts (larger than 35
$\mu g\ m^{-3}$ and less than 35 $\mu g\ m^{-3}$). Aeras with high pollution have $R^2$ of 0.64 and areas with low
pollution have $R^2$ of 0.67, therefore, the model performance is relative stable for both large and
small PM2.5 values. Also, the inclusion of low aerosol concentration areas does not influence the
model performance for high values (seen in supplemental material in igures S1 and S2), which
means that the high $R^2$ is not a reason of large number of low values. The GWR model fitting and
validation results for the 17 days in 2011 are shown in figure S3.
**4.3 Predictors' influence during wildfires**
Table 4 shows the GWR model mean coefficients for the whole US region and for different
selected regions. The selected boxes are shown in figure 4c in different colors: box1 (red) located
in NW US include major fire sources in US; box2 (gold) located in Montana state is influenced
from both neighboring states and  smoke from Canada; box3 (green) in Minnesota which is located
further from the fires and has minor increase in PM$_{2.5}$ due to remote smoke; box4 (black) in NE
(Northeast) US is the furthest from fires and has no obvious pollution increase. The second column
of the tables shows the conditions for sample selection and the third column shows the number of
pixels selected for each box. By comparing the coefficients of samples selected in these boxes,
predictors have different influence in different locations. AOD has stronger influence on predicting
PM$_{2.5}$ closer to fire sources, but local emissions become more dominant if the distances is large
enough. The smoke flag is overall positive related to surface PM$_{2.5}$, while it could slightly
negatively relate to PM$_{2.5}$ around fire sources and northeastern coasts. PBL is negatively related to
PM$_{2.5}$ when the pollution is concentrated near the surface (fires or human-made emissions), while
it appears to be positively related to PM$_{2.5}$ at locations where the main pollution source comes
from remote wildfire smoke. Surface temperature have a relative stable positive correlation with
surface PM$_{2.5}$, however, surface pressure and wind speeds are negatively correlated with PM$_{2.5}$.
Relative humidity, on the other hand, shows large variations on PM$_{2.5}$ influence across the nation.
Around the wildfires where the RH is relative low, RH has a positive correlation with PM$_{2.5}$ since
hygroscopicity would increase and leads to accumulation of PM$_{2.5}$, but increasing RH can also
decrease PM$_{2.5}$ concentration by overgrowing  the PM$_{2.5}$ particles to deposition at high RH
environment (Chen et al., 2018).

From table 4, we know that the weighting for AOD is much larger than other predictors, but

predictors other than AOD are important for the prediction. We tested our model with AOD as the only
predictor to conduct a comparison with the original model, and the R$^2$ decreases from 0.83 to 0.79 and
RMSE increases from 3.46 to 3.8. This is consistent with previous study (Jiang et al., 2017) which
shows improvements of R$^2$ from 0.69 to 0.78 and RMSE from 7.25 to 6.18 by adding 4 meteorological
parameters in summer in easter China. Other predictors have higher weighting at the fire source region
(box1) where BLH cannot provide the aerosol vertical distribution information since smoke tends to
be injected to higher levels. For high AOD regions where aerosol tends to be suspended at high levels,
adding other predictors other than AOD tends to have lower improvement of the model compared with
low AOD values, because adding BLH can significantly improve the prediction for low level aerosols.
For regions with AOD less than 35, R$^2$ increases 0.09 from AOD only model (0.6 to 0.69), while R$^2$
increases 0.05 for areas with AOD larger than 35. RMSE decreases 12% and 7% for AOD less and
larger than 35 conditions, respectively. Overall, the meteorological factors have larger improvements
for low polluted areas (low level aerosol in this case).

**4.4 Predicted PM$_{2.5}$ Distribution**

The mean PM$_{2.5}$ distributions over the United States shown in Figure 4a is calculated by

averaging the surface PM$_{2.5}$ data from ground monitors for the 17 days, which matches well with
the GWR model-predicted PM$_{2.5}$ distributions shown in Figure 4b. The model estimation extends
the ground measurements and provide pollution assessments across the entire nation. Comparing
the AOD map (Figure 2b) with the PM$_{2.5}$ estimations (Figure 4b), demonstrates the differences
between columnar and surface-level pollution. Differences between the AOD and PM$_{2.5}$
distributions are due to various reasons including 1) Areas with high $PM_{2.5}$ concentrations in figure
4b correspond to low AOD values in figure 2b (Southern California, Utah, and southern US); 2)
and high AOD regions in figure 2b correspond to low $PM_{2.5}$ concentrations in figure 4b
(Minnesota). The first situation usually occurs at the edge of polluted areas that are relative far
from the fire source, which is consistent with previous studies that reported smaller particles (<10
$\mu g$) are able to travel longer distances compared to large particles (>10 $\mu g$) (Gillies et al., 1996),
and that lager particles tend to settle closer to their source (Sapkota et al., 2005; Zhu et al., 2002).

We use the same method for August 9th to August 25th in 2011 that had low fire activity,

ensuring consistency for estimating coefficients for different variables for 2011. Figure 4c shows
the difference in spatial distribution of mean ground $PM_{2.5}$ of the 17 days between 2018 and 2011.
High values of $PM_{2.5}$ differences are in the Northwestern and central parts of the United States
with the Southern states having very little impact due to the fires. Of all the 48 states within the
study region, there are 29 states that have a higher $PM_{2.5}$ value in 2018 than 2011, and 15 states
have 2018 $PM_{2.5}$ value more than two times their 2011 value (shown in figure 5). The mean $PM_{2.5}$
for WA increases from 5.87 in 2011 to 46.47 $\mu g\ m^{-3}$ in 2018, which is about 8 times more than
2011 values. The $PM_{2.5}$ values in Oregon increases from 4.97 (in 2011) to 33.3 $\mu g\ m^{-3}$ in 2018,
which is nearly seven times more than in 2011. For states from Montana to Minnesota, the mean
$PM_{2.5}$ decreases from east to west, which reveals the path of smoke transport. As shown in Figure
4c, there is a clear transport path of smoke from North Dakota all the way to Texas. Along the
path, smoke increases $PM_{2.5}$ concentrations by 168% in North Dakota and 27% in Texas. Smoke
aerosols transported over long distances contains fine fraction PM which significantly affect the
health of children, adults, and vulnerable groups.

Figure 5 shows the mean PM2.5 predicted from the GWR model of different EPA regions

for the 17 days in 2011 and 2018 (Hawaii and Alaska are not included). The most influenced region

is region 10, which has a 2018 mean PM2.5 value of 34.2 $\mu g\ m^{-3}$ that is 6 times larger than the

values in 2011 (5.8 $\mu g\ m^{-3}$) values. The PM2.5 of region 8 and 9 have 2.4 and 2.6 times increase

in 2018 compared to 2011. Region 1~4 have lower PM2.5 in 2018 than 2011 possibly due to Clean

Air Act initiatives, absence of any major fire activites and further away for transported aerosols.

The emission reduction improves the US air quality and lower the PM2.5 every year, but 6 out of

10 EPA regions show significant increases in PM2.5 during the study period, which indicates that

the long-range transported wildfire smoke has become the new major pollutant in the US.

**4.5 Estimation of Canadian fire pollution**

To evaluate the pollution caused only from Canadian fires, we did a rough assessment

according to the total FRP and PM2.5 values. There are three states in the US have wildfires during

the study period: California, Washington and Oregon, and they have total FRP of 1186, 518 and

439 (*1000 MW) respectively. Assuming that California was only influenced by the local fires,

then fires of 1186 (*1000 MW) cause 13 $\mu g\ m^{-3}$ increase in PM2.5. Accordingly, wildfires in

Washington and Oregon State will cause 6 and 5 $\mu g\ m^{-3}$ increase in state mean PM2.5. Therefore,

Canadian fires caused PM2.5 increase in Washington and Oregon is about 35 and 23 $\mu g\ m^{-3}$. Since

the FRP of Canadian wildfires are approximately 5 times larger than that of the California fires,

which is the strongest fire in US, we assume the pollution affecting the states located in the

downwind directions other than the three states are mainly coming from Canadian wildfires. States

with no local fires such as Montana, North Dakota, South Dakota and Minnesota have PM2.5

increase of 18.31, 12.8, 10.4 and 10.13 $\mu g\ m^{-3}$. The decrease of these numbers reveal that the

smoke is transport in a SE direction. This influence of Canadian wildfires on US air quality is only

a rough quantity estimation, thus additional work is needed for understand long-range transport
smoke pollution and its impact on public health. One way to do this would be assessing the
difference of pollution by turning on and off US fires in chemistry models.
**4.6 Comparison with previous studies**
Comparing with the Bayesian ensemble model developed by Geng et al. (Geng et al., 2018)
using MAIAC AOD and CMAQ (Community Multiscale Air Quality) model and ground PM$_{2.5}$
measurements, our GWR model has larger R$^2$, but with the chemistry transport model (CTM),
their method can provide more vertical distribution information which is important for wildfire
smoke. GWR usually have better accuracy than CTM since there are large uncertainties related to
different CTM inputs such as emission, meteorological and land cover data, but for regions with
less or no ground measurements, CTM provide a great approach for estimating surface PM$_{2.5}$.
Other studies which used machine learning method to predict surface PM$_{2.5}$ have better
performance for long-term prediction rather than monthly estimation (Liang et al., 2020; Xiao et
al., 2018), but can better resolve complex relationship between different predictors than statistical
models (Geng et al., 2020). For wildfire events, the available data is much less than the long-term
aerosol analysis, so the performance of machine learning method could be less accurate compared
to long-term prediction. Our study also shows slightly larger R$^2$ compared to other GWR studies
(Hu et al., 2013; Ma et al., 2014; You et al., 2016b) due to the inclusion of more meteorological
and other related predictors.
**4.7 Model uncertainties and limitations**
There are various sources of uncertainties and limitations for studies that use satellite data
to estimate surface PM$_{2.5}$ concentrations. Since wildfires develop quickly it is important to have
continuous observations to capture the rapid changes. This study uses polar orbiting high-quality
satellite aerosol products, but the temporal evolution can only be estimated by geostationary data
sets. Although satellite observations have excellent spatial coverage, missing data due to cloud
cover is a limitation. As discussed in the paper, the prediction error (RMSE) of the model is
between $3{\sim}5\ \mu g\ m^{-3}$, while the RMSE increased for locations with high aerosol concentration.
This is partly due to lack of accurate vertical distribution information which is very important for
wildfire smoke. The GWR model is largely influenced by the distribution of ground stations, and
the prediction error will be different in different places due to unevenly distributed $PM_{2.5}$ stations.
For locations that have a dense ground-monitoring distribution, the prediction error will be low,
while the prediction error will be relative larger at other places with sparse surface stations.
Although there are obvious limitations, complementing surface data with satellite products and
meteorological and other ancillary information in a statistical model like the GWR has provided
robust results for estimating surface $PM_{2.5}$ from wildfires. We also note that we did not consider
some variables used in other studies such as NDVI, forest cover, vegetation type, industrial
density, visibility and chemical constituents of smoke particles  (Donkelaar et al., 2015; Hu et al.,
2013; You et al., 2015; Zou et al., 2016). Visibility mentioned in some studies may improve the
model performance, but unlike AOD, it has limited measurement across the nation, which will
restrict the applicability of training data. Another uncertainty comes from the 2011 wildfires which
we assumed to be zero fire events but there are actually few fire events in EPA region 6, 8, 9 and
10, and this will lead to underestimation of $PM_{2.5}$ increase due to 2018 fires in these regions.

One limitation of this study is that analysis based on 17-day mean values cannot capture

daily pollution variations, which is also very important for pollution estimation during rapid-
changing wildfire events. To extend this analysis to daily estimation, the cloud contaminations of
satellite observations become a major problem. Therefore, future work is needed using chemistry
transport models and other data to fill in the gaps on missing AOD data due to cloud coverage.

**5. Summary and Conclusions**

We estimate the surface mean $PM_{2.5}$ for 17 days in August for a high fire active year (2018)
and compare that with a low fire activity year using the Geographically Weighted Regression
(GWR) method to assess the increase in $PM_{2.5}$ in the United States due to smoke transported from
fires. The difference in $PM_{2.5}$ between the two years indicates that more than half of the US states
(29 states) are influenced by the NWUSC wildfires, and half of the affected states have 17-day
mean $PM_{2.5}$ increases larger than 100% of the baseline value. The peak $PM_{2.5}$ during the wildfires
can be much larger than the 17-day average and can affect vulnerable populations susceptible to
air pollution. Some of the most affected states are in Washington, California, Wisconsin, Colorado
and Oregon, all of which have populations greater than 4 million. According to CDC (Centers for
Disease Control and Prevention), 8% of the population have asthma (CDC, 2011). Therefore, for
asthma alone, there are about 3 million people facing significant health issue due to the long-range
transport smoke in these states.
For states that show decrease in $PM_{2.5}$ due to the Clean Air Act, the mean decrease is about
16% of the baseline after 7 years. This is consistent with EPA's report that there is a 23% decrease
of $PM_{2.5}$ in national average from 2010 to 2019(U.S. Environmental Protection Agency, 2019).
Comparing with the dramatic increase (132%) caused by wildfires, pollution from the fires is
counteracting our effort on emission controls. Although wildfires are often episodic and short-
term, high frequency of fire occurrence and increasing longer durations of summertime wildfires
in recent years has made them now a long-term influence on public lives. Our results show a
significant increase of pollution in a short time period in most of the US states due to the NWUSC
wildfires, which affects millions of people. With wildfires becoming more frequent during recent
years, more effort is needed to predict and warn the public about the long-range transported smoke
from wildfires.
**Acknowledgements.**
Pawan Gupta was supported by a NASA Grant. MODIS data were acquired from the Goddard
DAAC. Sincerest thanks to the MAIAC, MODIS, EPA and ECMWF teams for their datasets that
makes  this research possible.

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

Table 1. Datasets used in the study with sources.

|   | Data /Model | Sensor | Spatial Resolution | Temporal Resolution | Accuracy |
|---|---|---|---|---|---|
| 1 | Surface PM$_{2.5}$ | TEOM | Point data | daily | ±5~10% |
| 2 | Mid visible aerosol optical depth (AOD) | MAIAC_ MODIS | 1km | daily | 66% compared to AERONET |
| 3 | Fire Radiative Power (FRP) | Terra/Aqua-MODIS | 1km | daily | ± 7% |
| 4 | ECMWF (Meteorological variables) | | 0.25 degree | hourly | |
|   | | | | | |

1) https://www.epa.gov/outdoor-air-quality-data
2) https://earthdata.nasa.gov/
3) https://earthdata.nasa.gov/
4) https://www.ecmwf.int/en/forecasts



Table 2. Total FRP in Canada and Northwestern US in August of Different Years (unit: $10^4$

737                                    MW)

| Year | 2010 | 2011 | 2012 | 2013 | 2014 | 2015 | 2016 | 2017 | 2018 |
|------|------|------|------|------|------|------|------|------|------|
| CA | 148.24 | 4.84 | 19.93 | 70.54 | 107.78 | 10.39 | 4.6 | 307.3 | 542.99 |
| NW US | 16.41 | 42.84 | 320.39 | 192.06 | 67.01 | 339.58 | 112.9 | 195.64 | 296.91 |


Table 3. statistics of 15 states that violate EPA standards ($35\ \mu g\ m^{-3}$) during the 17-day wildfire
period

| State | number of site violate standard | number of site in the state | Percentage of site violate standard (%) | number of days violate standard |
|-------|-------|-------|-------|-------|
| Montana | 14 | 15 | 93.34 | 16 |
| Washington | 18 | 20 | 90 | 16 |
| Oregon | 12 | 14 | 85.71 | 5 |
| North Dakota | 7 | 11 | 63.63 | 4 |
| Idaho | 5 | 8 | 62.5 | 8 |
| Colorado | 11 | 21 | 52.38 | 2 |
| South Dakota | 5 | 10 | 50 | 1 |
| California | 57 | 119 | 47.9 | 14 |
| Utah | 7 | 15 | 46.67 | 4 |
| Nevada | 4 | 13 | 30.77 | 1 |
| Wyoming | 7 | 24 | 29.2 | 2 |
| Minnesota | 4 | 26 | 15.4 | 2 |
| Texas | 3 | 37 | 8.1 | 1 |
| Louisiana | 1 | 14 | 7.1 | 1 |
| Arizona | 1 | 20 | 5 | 1 |


Table 4. Coefficients of different predictors

| Mean coefficients | sample selection | N | AOD | smoke flag | PBL | T2M | RH | U | SP |
|-------|-------|-------|-------|-------|-------|-------|-------|-------|-------|
| box1(red) | FRP>1000 | 213 | 91.94 | -0.14 | -2.25 | 0.33 | 0.08 | -2 | -0.06 |
| box2(gold) | PM2.5>30 | 362 | 60.1 | 0.013 | -2.9 | 0.23 | -0.08 | -1.6 | -0.03 |
| box3(green) | PM2.5>17 | 278 | 6.2 | 0.05 | 0.2 | 0.2 | 0.014 | -0.3 | -0.02 |
| box4(black) | 17>PM2.5>10 | 938 | 7.1 | -0.02 | -1.2 | 0.22 | -0.035 | 0.06 | -0.005 |
| whole US region | ~ | 106352 | 28.1 | 0.024 | -0.9 | 0.06 | -0.04 | -0.7 | -0.002 |



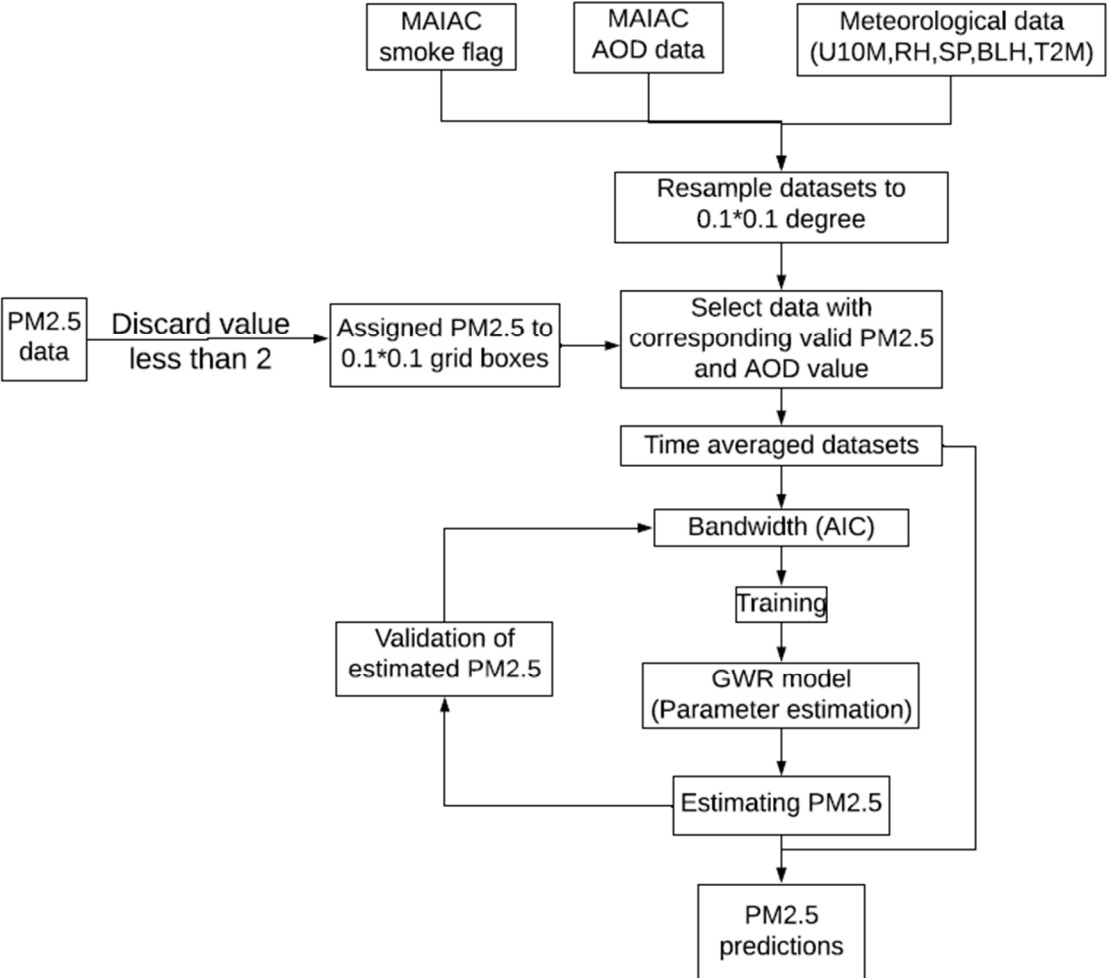


Figure 1. Flow chart for the Geographically Weighted Regression model used. All satellite,

ground, meteorological data are gridded to 0.1 by 0.1 degrees.



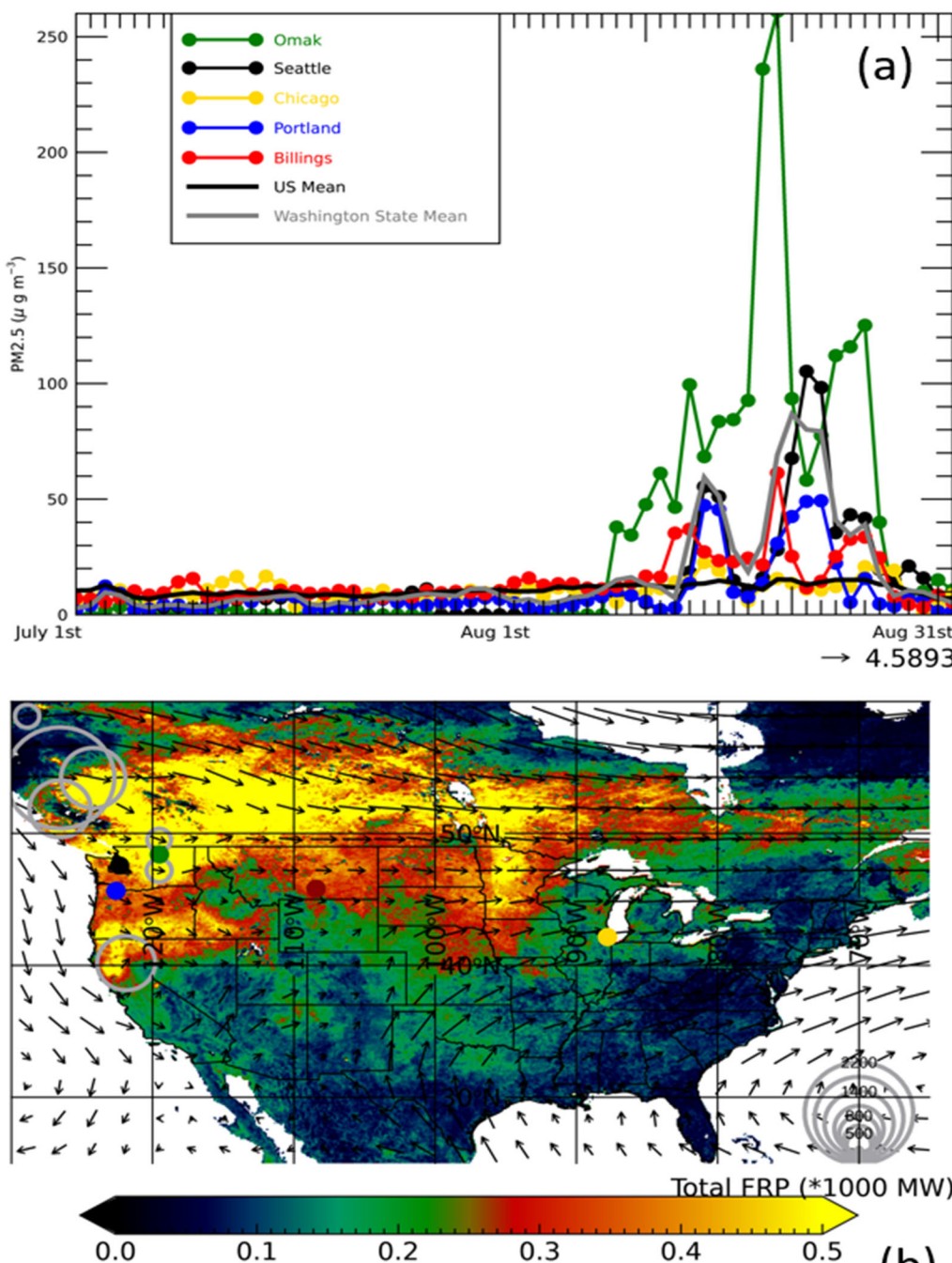

Figure 2. (a) Variations of EPA ground observed PM2.5 in different cities from July to August 2018 (Omak-Washington, Seattle-Washington, Chicago-Illinois, Portland-Oregon, Billings-Montana). Black line without markers shows the mean variation of the whole US stations and the grey line without markers shows the mean variation of stations in Washington state. (b) Mean MAIAC satellite AOD distribution from August 9th to August 25th, 2018. AOD values equal or larger than 0.5 are shown as the same color (yellow). Also shown are circles with Fire Radiative Power (FRP). Black arrow shows the wind direction and the length of it represents the wind speed. The round spots of different colors on the map show the locations of the five selected cities (green-Omak, black-Seattle, yellow-Chicago, blue-Portland, red-Billings).

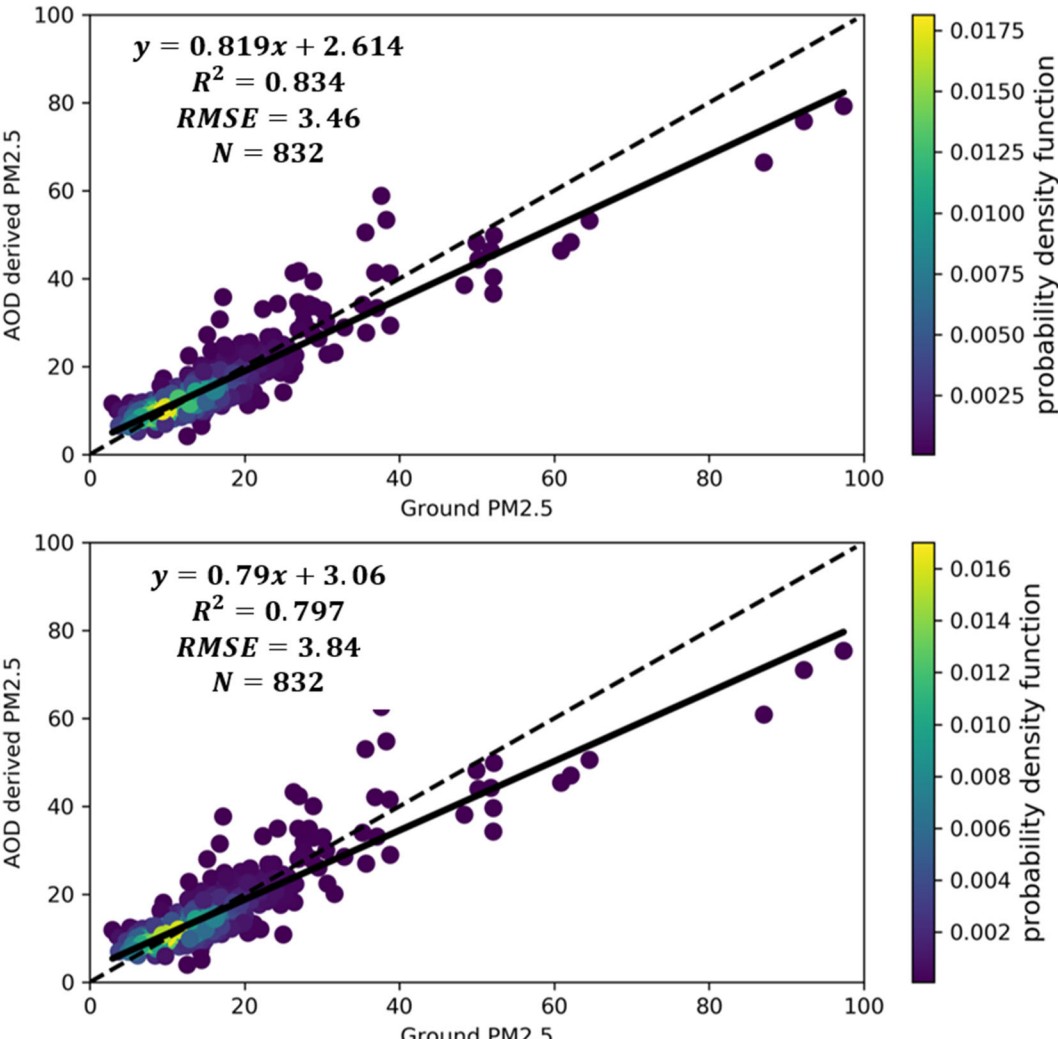

Figure 3. Results of model fitting and cross validation for GWR model for the entire US region averaged from August 9th to August 25th, 2018. (a) GWR model fitting results (b) GWR model LOOCV results. The dash line is the 1:1 line as reference and the black line shows the regression line. The color of the scatter plots represents the probability density function which provides a relative likelihood that the value of the random variable would equal a certain sample.


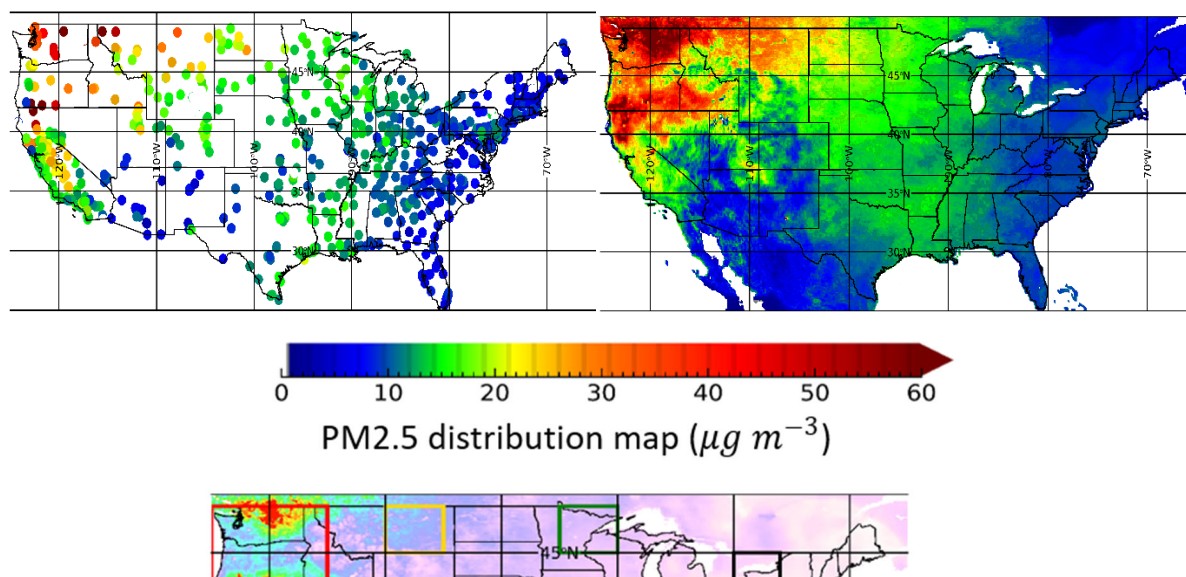


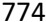


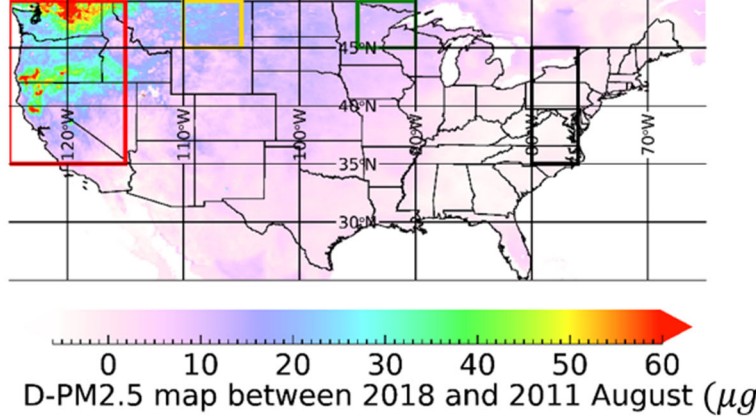


Figure 4. (a) EPA ground observed PM$_{2.5}$ distribution over the US averaged from August 9th to
August 25th, 2018. (b) GWR predicted 17-day mean PM$_{2.5}$ distribution. (c) Difference map of
predicted ground PM$_{2.5}$ of the 17-day mean values between 2018 and 2011. PM$_{2.5}$ values equal or
larger than 60 $\mu g\ m^{-3}$ are shown as the same color (red). Note that the D-PM$_{2.5}$ has a different
color scale to make the negative values more apparent (blue).


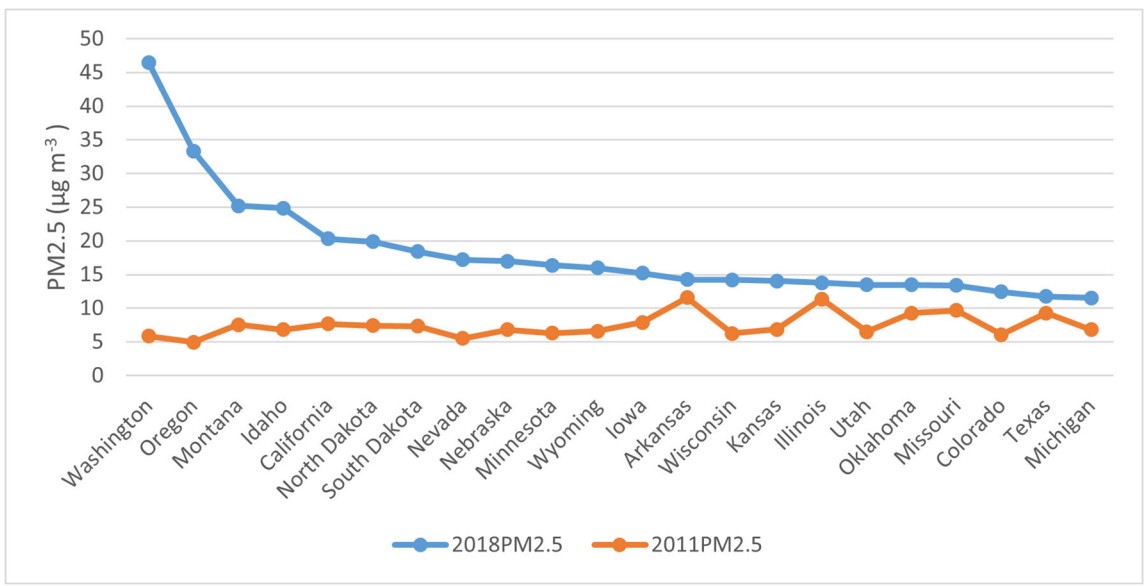


Figure 5. Mean PM$_{2.5}$ from August 9$^{th}$ to August 25$^{th}$ in 2018 and 2011 of most affected states


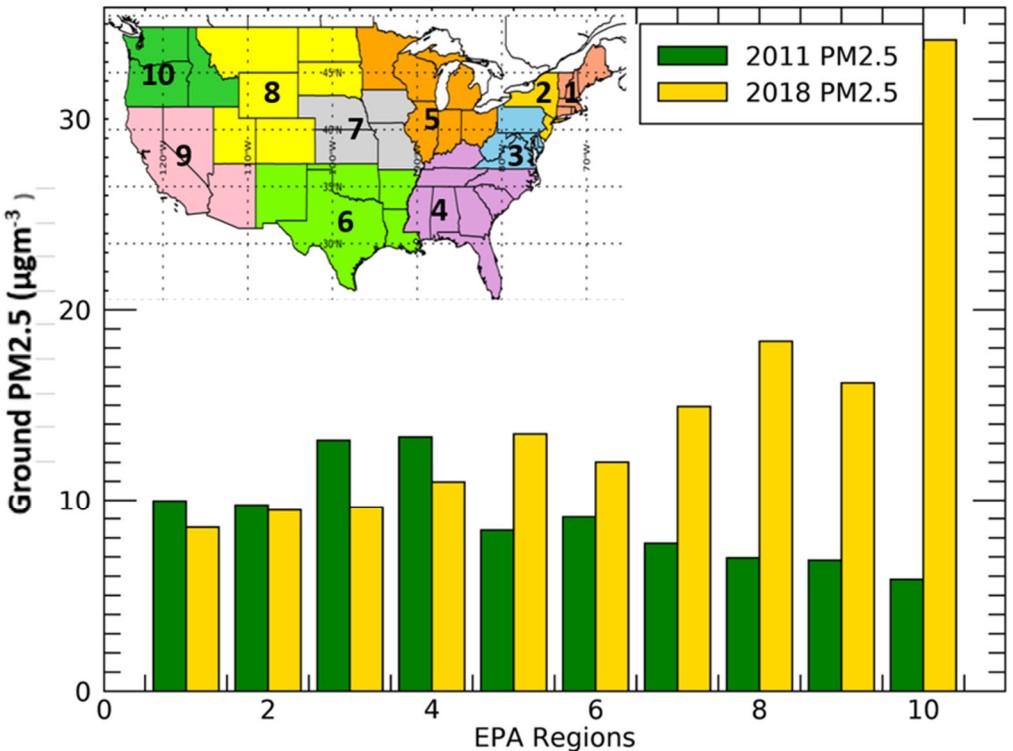


Figure 6. Mean $PM_{2.5}$ of EPA regions from August 9th to August 25th in 2011 and 2018. Inset
shows the map of 10 EPA regions in different colors. Yellow column represents the 2018 mean
$PM_{2.5}$ and green column represents for 2011 mean $PM_{2.5}$.
