# Peer review of "1. Introduction"

_Atmospheric Chemistry and Physics, 2020_

## Referee Comment (RC1) · Anonymous Referee #1 · 4 Jan 2021

Review of the paper "Satellite-based Estimation of the Impacts of Summertime Wildfires on Particulate Matter Air Quality in United States" by Xue et al. to ACP

This paper uses long-term Aqua and Terra MODIS Fire Radiative Power (FRP), Aerosol Optical Depth (AOD), and surface observations of PM2.5 (particulate matter with median diameter smaller than 2.5 um) to study the impact of smoke from wildfires on surface PM2.5. The authors picked a 2-week time period in 2018 to represent extreme wildfires and 2011 to represent low wildfire activity to compare and contrast and report the impact. While it is true that the human induced pollution levels are going down in the US and the role of natural events such as wildfires and dust storms in influencing the air quality is increasing, I find this work very rudimentary and without scientific rigor. The paper is well written no doubt but this work is merely an exercise of downloading data from different sources and making figures. Let me explain why I think this study needs a major re-work (scientific scope as well as methodology) and is not ready for publication: • First and foremost, to conduct this study, there is no need to use satellite data because the analysis is done in an aggregate sense, spatially and temporally. There are enough ground monitors in different states influenced by smoke from fires that a study can be designed just around the surface monitors without even bringing in the errors associated with scaling satellite AOD to surface PM2.5; • Second, some of the stations (100s of them) have daily (or every other day) speciation measurements including organic carbon and K+, biomarkers for smoke. The authors have not bothered to analyze the surface data and extract only data for the days or locations influenced by smoke. Yes, there are speciation observations from EPA network as well as interagency network (IMPROVE) in many of the states where smoke originates and many states downwind of smoke; • Third, if one or more ground monitors in a county/state are influenced by upwind smoke from fires, an exceptional events waiver must have been filed with the EPA. Did the authors check to see how many exceptional events waivers were filed for 2018 by the states that were under the smoke influence as reported by the authors? • Given that there are fire observations (ground reports from EPA as well as from satellites) and surface PM2.5 data for two decades, why not conduct or extend the study to all years to understand the nuances of the inter-annual variability and the influence of transport etc. Again, this is why I find this paper very premature because the authors have not even scratched the surface of the problem. It is indeed premature to talk about smoke particles (from satellites as well as ground observations) without bothering to understand if PM2.5 observed is indeed due to smoke or not; • There are many documented algorithms that use satellite data to flag smoke and smoke height including the MAIAC aerosol algorithm used in this study. The authors used AOD but not smoke flag and smoke plume height product generated by the same algorithm. While the smoke plume height product is

new, the smoke flag and AOD in the MAIAC algorithm are internally consistent and the authors should have used it in this study. Also there is no discussion on the quality of the MAIAC AOD and its performance. The algorithm performance is reported as 66% of the retrievals are within ±0.5? I am not exactly sure why this is a good performance? How good is the AOD product in different AOD ranges? Does it report AODs as high as 5 or 7 for these smoke events or smoke is misidentified as cloud? If an aerosol model is used in the algorithm, does the algorithm dynamically (correctly) pick smoke model for this time period? How consistently does it pick the smoke model? If another model is picked, what is the AOD bias for incorrectly picking a non-smoke model? And how does that translate to PM2.5 estimation error?

Regarding the GWR method and surface PM2.5 estimates:

• The authors indicate that the GWR is a proven method that is used by many but I have several questions. (1) Please show a map of regression parameters and demonstrate that the values have physical meaning, (2) give details on why you chose the parameters you chose for the model. Let us talk about population density. Why did you use it? I can understand why it is used if you are developing models for urban/industrial pollution where population density can be a proxy for traffic emissions etc. Here, isn't the focus of the study to understand the influence of long-range transport of smoke from fires on humans and their health. Then how can population density be a predictor? (3) no details given on the influence of different predictors such as boundary layer height on the prediction; • The authors have not shown their assessments on how good the estimated PM2.5 values are outside of one evaluation (scatter plot for the whole US). If you look at the density of the data points, most points are within 0 to 20 $\mu$g/m3 or so. When the EPA PM2.5 daily average standard is 35 $\mu$g/m3, I would be more interested in knowing the performance of the statistical model for exceedances. Can the authors actually tabulate what percentage of each jurisdiction (e.g., state) violated the daily standard and how many times within the 2-week window in 2018? Without metrics like that, the study really does not provide any value. • This work

also needs other corroborative evidence such as back trajectory cluster analysis to show the source-receptor relationship, analysis of LIDAR data (satellite or ground) to show evidence of transported smoke reaching the surface etc.

In summary, I cannot support the publication of this paper without new work carried out to address this problem in a more comprehensive way. Because this work has profound implications for air quality monitoring and policy, and the rigor is missing in designing a study to address the question of smoke influence on PM2.5, I have to unfortunately reject this paper.

---

## Referee Comment (RC2) · Anonymous Referee #2 · 1 Feb 2021

In the manuscript, the authors compared the satellite-derived PM2.5 in two different periods to see the impacts of wildfires on air quality in the US. Although the study presented some valuable results, it is relatively simple which lacks in-depth analysis, and the scientific innovation is not clear. In addition, I am mainly concerned about the used method for PM2.5 retrieval, and also a lot of important information is missing. Below are my specific comments: Line 54-70: The authors should carefully summarize the methods of PM2.5 estimations according to different categories, and the cited reference is too old and need to be updated by adding more recent studies. Line 86: The authors need clearly clarify the novelty of the study and the difference with previous related studies. Line 107: What's the accuracy of MAIAC AOD products in your study

region? I suggest adding a preliminary validation by comparing the AERONET ground-based measurements. Line 109-110: How do the authors deal with such a big cloud missing situation in such a short study period in summer? In this way, ground-based observations could be more suitable than satellite retrievals due to a large number of missing data. In addition, cloud and smoke are difficult to be distinguished during the AOD retrieval, resulting in the smoke areas are often masked as clouds? Line 117: Why not use the ERA5-Land meteorological data at a finer resolution of 0.1 degrees? Line 146: 0.1° or 0.01°? MAIAC AOD is 1 km. Section 3.3: The reviewer doesn't know why the authors choose the GWR model since there are many existed more accurate statistical regression (e.g., GTWR) or machine learning (e.g., random forest) models that have been proved in previous studies. The author should clearly clarify this. Line 177: What is the LOOCV method and how does it work? Table 2: Should be improved (a line or bar chart might be better), in addition, state abbreviations are hard to read. The result analysis is very simple, which seems like an article about the PM2.5 retrieval algorithm. More in-depth analysis of the impacts of wildfires on air quality is needed.

---

## Author Comment (AC1) · 3 Mar 2021

Response to reviewer

This paper uses long-term Aqua and Terra MODIS Fire Radiative Power (FRP), Aerosol Optical Depth (AOD), and surface observations of PM2.5 (particulate matter with median diameter smaller than 2.5 um) to study the impact of smoke from wildfires on surface PM2.5. The authors picked a 2-week time period in 2018 to represent extreme wildfires and 2011 to represent low wildfire activity to compare and contrast and report the impact. While it is true that the human induced pollution levels are going down in the US and the role of natural events such as wildfires and dust storms in influencing the air quality is increasing, I find this work very rudimentary and without scientific rigor. The paper is well written no doubt but this work is merely an exercise of downloading data from different sources and making figures. Let me explain why I think this study needs a major re-work (scientific scope as well as methodology) and is not ready for publication.

We thank the reviewer for the extensive and detailed review of our manuscript. We believe the comments improved the paper, and we have revised the paper significantly in light of the reviewer's suggestions. Our point-by-point response to the reviewer's comments is given below:

First and foremost, to conduct this study, there is no need to use satellite data because the analysis is done in an aggregate sense, spatially and temporally. There are enough ground monitors in different states influenced by smoke from fires that a study can be designed just around the surface monitors without even bringing in the errors associated with scaling satellite AOD to surface PM2.5;

While there would be less errors without using satellite AOD, pollution cannot be estimated at places that lack ground monitors such as Wyoming (shown in figure 1 below). We also added some explanation in the discussion section.

Canadian wildfires in some years affected the US east coast, and the smoke can be captured by ground monitors. However, wildfires in summer 2018 mostly influenced the northern and western part of US where, unlike eastern coast, population is less. Ground monitors of many affected states are only concentrated in the few population centers and leave large gaps of PM2.5 observations in these states. For example, there are 13 EPA PM2.5 stations in Wyoming state (figure 1a), but they are distributed in two corners, while leaving a large portion of the state unmonitored. It is similar a situation in other states except for Washington and California State.

It is possible to estimate PM2.5 by applying interpolation methods on the ground observations but not feasible when there are large gaps. Also we decided to use GWR method and utilize different variables to increase the prediction accuracy.

[Figure]

Figure 1. 17-day mean PM2.5 distribution from (a) EPA ground monitors (b) GWR estimation

Second, some of the stations (100s of them) have daily (or every other day) speciation measurements including organic carbon and K+, biomarkers for smoke. The authors have not bothered to analyze the surface data and extract only data for the days or locations influenced by smoke. Yes, there are speciation observations from EPA network as well as interagency network (IMPROVE) in many of the states where smoke originates and many states downwind of smoke;

[Figure]

Figure 2. IMPROVE data.

We thank the reviewer for pointing this out. We actually checked selected ground observations, and assessed that high PM2.5 indeed originated from fires. We did not intend to filter the data to analyze only for smoke pixels since an estimation of smoke affected regions also needs unaffected pixels to feed the model.

We checked a few IMPROVE sites of the affected states, and found high organic carbon mass concentration and noticeable increases in elemental carbon, ammonium nitrate and ammonium sulfate. Figure 2 is an example of one site in Montana State (115.6709°W, 47.9549°N). We also compared the EPA PM2.5 distributions with other species from the EPA stations (Figure 3), and we conclude that the increase of PM2.5 is due to wildfires since high values of PM2.5, elemental carbon PM2.5 and organic carbon PM2.5 all distributed the same.

[Figure]

Figure 3. Surface Distribution of chemical speciation

Third, if one or more ground monitors in a county/state are influenced by upwind smoke from fires, an exceptional events waiver must have been filed with the EPA. Did the authors check to see how many exceptional events waivers were filed for 2018 by the states that were under the smoke influence as reported by the authors?

We checked the exceptional events waivers for 2018 (Figure 4). The results are consistent with our findings: the affected states from our analysis had indeed more exceptional events waivers.

[Figure]

Figure 4. Exceptional Events

Figure 4 shows the number of exceptional events waivers filed during the study period (August 9th to August 25th, 2018) in different states. States with no exceptional events waivers during this period are not shown in the figure. The values are not only related to the number of EPA sites in each states but also relevant to the distribution (location) of these sites.

Given that there are fire observations (ground reports from EPA as well as from satellites) and surface PM2.5 data for two decades, why not conduct or extend the study to all years to understand the nuances of the inter-annual variability and the influence of transport etc. Again, this is why I find this paper very premature because the authors have not even scratched the surface of the problem.

It would have been interesting to explore this aspect. However, the purpose of this paper is to apply GWR method on a selected severe wildfire event, test its predicting performance over region that contains high concentration smoke, and then quantify the influence of one wildfire event on the US air quality. Studying the variability of past 20 years' wildfires is beyond the scope of this paper.

There are many documented algorithms that use satellite data to flag smoke and smoke height including the MAIAC aerosol algorithm used in this study. The authors used AOD but not smoke flag and smoke plume height product generated by the same algorithm. While the smoke plume height product is new, the smoke flag and AOD in the MAIAC algorithm are internally consistent and the authors should have used it in this study. Also there is no discussion on the quality of the MAIAC AOD and its performance. The algorithm performance is reported as 66% of the retrievals are within 0.5? I am not exactly sure why this is a good performance? How good is the AOD product in different AOD ranges? Does it report AODs as high as 5 or 7 for these smoke events or smoke is misidentified as cloud? If an aerosol model is used in the algorithm, does the algorithm

dynamically (correctly) pick smoke model for this time period? How consistently does it pick the smoke model? If another model is picked, what is the AOD bias for incorrectly picking a non-smoke model? And how does that translate to PM2.5 estimation error?

The smoke flag is added to the GWR model as a predictor and we have revised the reference for MAIAC AOD performance. Over North America, MAIAC AOD has a very small bias of -0.01 compared to AERONET AOD (Superczynski et al., 2017). The typical error is usually around ±0.05 during times of high aerosol loading, and the bias slightly increases as AOD increases.

The MAIAC AOD product has a maximum value of 4, which should be enough for our study since the major fire sources in Canada are far away from the US. The smoke detection is performed using MODIS red, blue and deep blue bands, and separate smoke pixels from dust and clouds based on absorption parameter, size parameter and thermal threshold. For now, there is no explicit biomass burning aerosol models included in the MAIAC retrievals. For pixels with no smoke detected, upper 50% of the data will be filtered as potentially affected by clouds or shadows, which will possibly lead to missing data.

Please show a map of regression parameters and demonstrate that the values have physical meaning

The figures below show the distribution of some regression coefficients. AOD coefficients are greater close to the fire sources (Northwestern US) and gradually decreases with distances increase, which means AOD is more dominant in predicting PM2.5 near the fire.

The smoke flag is overall positive related to surface PM2.5, while it could slightly negatively relate to PM2.5 around fire sources and northeastern coasts.

The PBL are negatively related to PM2.5 when the pollution is concentrated around the surface (fires or human-made emissions), while it appears to be positive related to PM2.5 at locations where the main pollution source comes from remote wildfire smoke.

Relative humidity, on the other hand, shows large variations on PM2.5 influence across the nation. Around the wildfires where the RH is relative low, RH has a positive correlation with PM2.5 since hygroscopic would increase and leads to accumulation of PM2.5, but increasing RH can also decrease PM2.5 concentration by overgrowing the PM2.5 particles to deposition at high RH environment.

[Figure]

[Figure]

[Figure]

[Figure]

give details on why you chose the parameters you chose for the model. Let us talk about population density. Why did you use it? I can understand why it is used if you are developing models for urban/industrial pollution where population density can be a proxy for traffic emissions etc. Here, isn't the focus of the study to understand the influence of long-range transport of smoke from fires on humans and their health. Then how can population density be a predictor?

The reviewer is correct, and we have removed population density from the GWR model. More details on predictor choosing is described in Data section.

no details given on the influence of different predictors such as boundary layer height on the prediction

We have added some explanation on how predictors can influence the PM2.5 based on the coefficient's distribution (section 4.3).

The authors have not shown their assessments on how good the estimated PM2.5 values are outside of one evaluation (scatter plot for the whole US). If you look at the density of the data points, most points are within 0 to 20 ug/m3 or so. When the EPA PM2.5 daily average standard is 35 g/m3, I would be more interested in knowing the performance of the statistical model for exceedances. Can the authors actually tabulate what percentage of each jurisdiction (e.g., state) violated the daily standard and how many times within the 2-week window in 2018?

We think this is an excellent suggestion. For data greater than 35 $\mu g\ m^{-3}$, the model has a RMSE of 12.07 $\mu g\ m^{-3}$, which is a lot larger than the whole model RMSE. Therefore, the model has a tendency for underestimating PM2.5 exceedances by around 12.07 $\mu g\ m^{-3}$. The larger the PM2.5 is, the greater the model underestimates.

Also, in our study, it would not be possible to calculate the number of days that violating the EPA standard, because we estimate surface PM2.5 over a 17-day period, not daily estimation. But we add some analysis (with below table) using ground observations in the discussion section.

| state | number of site violate standard | number of site in the state | Percentage of site violate standard (%) | number of days violate standard |
|---|---|---|---|---|
| Montana | 14 | 15 | 93.34 | 16 |
| Washington | 18 | 20 | 90 | 16 |

| | | | | |
|---|---|---|---|---|
| Oregon | 12 | 14 | 85.71 | 5 |
| North Dakota | 7 | 11 | 63.63 | 4 |
| Idaho | 5 | 8 | 62.5 | 8 |
| Colorado | 11 | 21 | 52.38 | 2 |
| South Dakota | 5 | 10 | 50 | 1 |
| California | 57 | 119 | 47.9 | 14 |
| Utah | 7 | 15 | 46.67 | 4 |
| Nevada | 4 | 13 | 30.77 | 1 |
| Wyoming | 7 | 24 | 29.2 | 2 |
| Minnesota | 4 | 26 | 15.4 | 2 |
| Texas | 3 | 37 | 8.1 | 1 |
| Louisiana | 1 | 14 | 7.1 | 1 |
| Arizona | 1 | 20 | 5 | 1 |

this study also needs other corroborative evidence such as back trajectory cluster analysis to show the source-receptor relationship, analysis of LIDAR data (satellite or ground) to show evidence of transported smoke reaching the surface etc.

Based on the reviewer's suggestion, and we checked for several different datasets for the existence of smoke reaching surface and back-trajectory paths to find the smoke is indeed originated from both local and remote fire sources.

Below Figure shows the vertical feature mask from CALIPSO on August 19[th] 2018, and the blue line in the inset map on the top right corner represent for the satellite orbit track. Within US (shown in the red square box), there are large load of aerosols below 5km along the track (Idaho, Utah and Arizona).

The second figure shows the pollution (smoke) at Billings (Montana) on August 19[th] was originated from both remote fires in Canada and local fires in Washington and Idaho.

[Figure]

**Vertical Feature Mask** UTC: 2018-08-19 20:44:54.5 to 2018-08-19 20:58:23.2 Version: 4.20 Standard Daytime

| Lat | 19.95 | 26.05 | 32.14 | 38.20 | 44.25 | 50.26 | 56.22 | 62.11 | 67.84 |
| Lon | -108.01 | -109.46 | -111.03 | -112.77 | -114.74 | -117.06 | -119.93 | -123.69 | -129.00 |

1 = clear air    2 = cloud    3 = tropospheric aerosol    4 = stratospheric aerosol    5 = surface    6 = subsurface    7 = totally attenuated    L = low/no confidence

**NOAA HYSPLIT MODEL - TRAJECTORY FREQUENCIES**
**endpts per grid sq./# trajectories (%)    0 m and 99999 m**
Integrated from 2300 19 Aug to 0500 16 Aug 18 (UTC) [backward]
Freq Calculation started at 0000 00    00 (UTC)

[Figure]

**METEOROLOGICAL DATA**

Job ID: 189358                Job Start: Wed Feb 10 23:14:30 UTC 2021
Source 1    lat.: 45.80    lon.: -108.53    height: 500 m AGL
Initial trajectory started: 2300Z 19 Aug 18
Direction of trajectories: Backward      Trajectory Duration: 48 hrs
Frequency grid resolution: 0.50 x 0.50 degrees
Endpoint output frequency: 60 per hour
Number of trajectories used for this calculation: 8
Meteorology: 0000Z 19 Aug 2018 - NAM12

---

## Author Comment (AC2) · 3 Mar 2021

Response to review

In the manuscript, the authors compared the satellite-derived PM2.5 in two different periods to see the impacts of wildfires on air quality in the US. Although the study presented some valuable results, it is relatively simple which lacks in-depth analysis, and the scientific innovation is not clear. In addition, I am mainly concerned about the used method for PM2.5 retrievals, and also a lot of important information is missing.

We thank the reviewer for the helpful comments. The suggestions given by the reviewer helped to clarify our arguments and to improve the quality of the paper. We have made significant revisions to the paper based on comments from all reviewers.

Below are my specific comments: Line 54-70: The authors should carefully summarize the methods of PM2.5 estimations according to different categories, and the cited reference is too old and need to be updated by adding more recent studies.

As suggested by the reviewer, we added more recent references in the introduction section.

Line 86: The authors need clearly clarify the novelty of the study and the difference with previous related studies.

This study is novel in: 1) applying PM2.5 estimation methods on wildfire events and calculate the prediction error at high pollution concentration condition; 2) analyzing predictors' different influences in estimating PM2.5 under various conditions; 3) quantify the air pollution from fires by states and EPA regions.

Line 107: What's the accuracy of MAIAC AOD products in your study region? I suggest adding a preliminary validation by comparing the AERONET groundbased measurements.

We have revised the reference for MAIAC AOD performance. Over North America, MAIAC AOD has a very small bias of -0.01 compared to AERONET AOD (Superczynski et al., 2017). The typical error is usually around ±0.05 during times of high aerosol loading, and the bias slightly increases as AOD increases.

Line 109-110: How do the authors deal with such a big cloud missing situation in such a short study period in summer? In this way, ground-based observations could be more suitable than satellite retrievals due to a large number of missing data. In addition, cloud and smoke are difficult to be distinguished during the AOD retrieval, resulting in the smoke areas are often masked as clouds?

Cloud contamination is indeed an important limitation on estimating surface PM2.5 using satellite data, and that is also the reason for performing GWR in an aggregate sense in this study.

By aggregating satellite data for 17-days, we are able to predict PM2.5 using a reasonable amount of satellite observations. We also added one paragraph in model uncertainties and limitations section to provide one possible solution for these problems (both missing AOD data due to clouds and smoke misidentification). We plan to use chemistry transport models to fill in all the AOD gaps in the future work.

Line 117: Why not use the ERA5-Land meteorological data at a finer resolution of 0.1 degrees?

We agree with the reviewer that finer resolution of ERA-5 would be better for analysis, but ERA5-Land meteorological data does not have boundary layer height, which is very important for assessing surface PM2.5.

Line 146: 0.1∘ or 0.01∘? MAIAC AOD is 1 km.

The surface PM2.5 data we generate is 0.1-degree resolution due to the coarse resolution of meteorological datasets.

Section 3.3: The reviewer doesn't know why the authors choose the GWR model since there are many existed more accurate statistical regression (e.g., GTWR) or machine learning (e.g., random forest) models that have been proved in previous studies. The author should clearly clarify this.

We agree that other statistical regression especially GTWR would improve the accuracy on PM2.5 predictions. However, the limitation on AOD data (missing data due to cloud and other gaps) is one problem for conducting the model since it needs more training data. This is also the reason for conducting this analysis on a 17-day aggregate sense.

Line 177: What is the LOOCV method and how does it work?

LOOCV is actually an extreme version of k-fold cross validation, and it requires maximum computational cost because, for each data in the dataset, we will create one model and evaluate this data. Therefore, the results are reliable and unbiased though computationally expansive.

Table 2: Should be improved (a line or bar chart might be better), in addition, state abbreviations are hard to read. The result analysis is very simple, which seems like an article about the PM2.5 retrieval algorithm. More in-depth analysis of the impacts of wildfires on air quality is needed.

Thank you for pointing this out. We changed the state abbreviations to full names and added a plot to show only some most influenced states. We also add more analysis in the discussion section.

---

## Referee Report (RR1)

Review of Xue et al., Satellite-based Estimation of the Impacts of Summertime Wildfires on Particulate Matter Air Quality in United States

This study uses the GWR method to predict surface PM2.5 concentrations in the US based on satellite AOD and meteorological variables. The statistic method is robust and is well referenced from previous studies, and the prediction results show good agreement with the in-situ measurements. However, the study is still lack of adequate scientific expansion from the results, and the conclusions are similar to the studies on satellite AOD products or ground measurements only, making this study less meaningful.

Before the consideration of publishing, the authors need to further explore the prediction results, make good findings or quantifications that simple AOD and scattering ground measurements cannot show. The authors also need to clean up the minor typos, formats, the potential figure-caption disagreements and misleading journal names in the page head.

The major difference between the science of AOD and surface PM2.5 is that, AOD is showing the vertical column conditions instead of surface only. Even not considering the aerosol chemistry and secondary formation in clouds, the convection conditions, atmospheric stability or vertical profiles of other meteorological conditions should contribute very much to the difference of AOD and surface PM. Especially for fire plumes, the long-term transport of fire smoke can be at a high altitude, and the vertical pattern of PM will be very different from the no-fire patterns. However, in the GWR model used in this study, only near-surface data are used. Also, noticing the AOD coefficient is much higher that all the other predictors (Table 4). It is doubtful how good the model is, compared to the agreement between AOD and surface PM. Therefore, the authors need to:

1. Show the improvement of the model from using AOD as the only factor, and discuss how the model predict the surface PM out of a column variable.
2. Estimate the model performance only looking at fire region, compare to Figure 3, and discuss the performance and potential bias.

Except for the main concerns, there are some minor suggestions and questions listed below:

1. Line 141-143: Since all the regions in the US are evaluated (in Figure 6), FRP in the other regions should also be verified, to make sure the 2011-2018 difference over the regions other than NW US is not affected by regional fire.
2. Line 269-270: as discussed in 1.2, further discussions e.g. calculation $R^2$ for high PM values may be useful.

3. Line 274-278: Is there any logic about the box selecting? For example, how to decide the size of the box? Can the box be larger? For each type of region, the authors can also show a regional mean with standard deviations of each coefficient. Also, can the box/region selection be more quantified, for example, by classifying using the background PM concentrations or FRP?
4. Line 612-614: The caption of Figure 4 seems not agree with the figure it self. "*PM2.5 values equal or larger than 30 μg m-3 are shown as the same color (red)...*", but the color label is ~-5 to 60 ug/m3.

---

## Referee Report (RR2)

**Second review of Xue et al., Satellite-based Estimation of the Impacts of Summertime Wildfires on Particulate Matter Air Quality in United States**

The authors did not address the major issue I pointed out in the first response, and the revised manuscript did not add any information supporting a deeper finding or importance of this work. Some of the questions are well answered, which should be good for the scientific significance, but the corresponding updates did not appear in the revised manuscript. The authors need to further consider this issue and do full discussions in the manuscript.

Before discussing the old issue: although it may not be a reviewer's responsibility, the font size in Section 4.3 is smaller than the previous test, and the acknowledgment did not show credits to the in-situ $PM_{2.5}$ measurements or the ECMWF products. Keeping these errors and typos will hurt the credibility of the journal.

1. For the response to my major issue 1, the response is not describing how the column AOD is related to $PM_{2.5}$. Here is the reason:

Here shows the equation (1) the authors added:

$$AOD = PM_{2.5}\, H\, f(RH)\, \frac{3 Q_{ext,dry}}{4\rho\, r_{eff}} = PM_{2.5}\, H\, S$$

This equation represents the AOD in a specific layer. If 'H' is the height of the column, the 'AOD' is the column AOD; if 'H' is the PBLH, then the 'AOD' on the LHS is just an AOD of the PBL. Thus, when in the response the authors claimed that H represents PBLH, the AOD is not the total column from satellite data. This claim did not appear in the manuscript. Therefore, the main issue pointed out in the last review was not solved. AOD does have high correlation with $PM_{2.5}$ because usually the vertical $PM_{2.5}$ profile does not significantly change, and the surface source of $PM_{2.5}$ is mostly mixed in the BL. It is right in most cases, but not in the fire cases. The average fire plume height is 1-2 km, and sometimes the plume can go to 5 km, or even higher. When this study is focusing on wildfires, this issue cannot be ignored, and the free-troposphere transport of smoke will be a robust bias in the model, because the model did not include any parameters with the vertical information. This probably leads to the underestimation of the prediction of the large values. The authors need to well address and discuss this issue in the manuscript.

The second part of this response discussing the previous studies is good for stating the significance of the model and the study, which also answers another reviewer's question about the correlation with meteorological data, but it seems not in the manuscript. This also helps the discussion in the previous paragraph: when the vertical profile information of PM is missing in the model, the weight of parameters other than AOD should be higher. The authors need to discuss this in the manuscript with Table 4, showing the agreement or disagreement about the AOD weight in this study and previous studies.

2. The figures showing the smoke impact region and NW US addressed my question, but not showing in the manuscript. If the authors prefer a clear main text, I recommend the authors include this in the supplementary materials, because without it, it will be the reader's concern that the low values from regions free from fire smoke may dominates the high R.

3. In addition to the previous review and response, the authors need to compare the method and model performance with previous studies. Some examples include:

Liang, F., Xiao, Q., Huang, K., Yang, X., Liu, F., Li, J., Lu, X., Liu, Y. and Gu, D., 2020. The 17-y spatiotemporal trend of PM2. 5 and its mortality burden in China. Proceedings of the National Academy of Sciences, 117(41), pp.25601-25608.
Xiao, Q., Chang, H.H., Geng, G. and Liu, Y., 2018. An ensemble machine-learning model to predict historical PM2. 5 concentrations in China from satellite data. Environmental science & technology, 52(22), pp.13260-13269.
Geng, G., Meng, X., He, K. and Liu, Y., 2020. Random forest models for PM2. 5 speciation concentrations using MISR fractional AODs. Environmental Research Letters, 15(3), p.034056.

I am not listing all, there are a lot of $PM_{2.5}$ estimations from AOD in the US and around the world. The authors need a full literature review to estimate the advantage and disadvantage of the model methodology, the model performance compared to previous studies, and the performance applying in fire.

Also, previous studies of fire $PM_{2.5}$ estimates such as Geng et al (2018) and else also need to be discussed.

Geng, G., Murray, N.L., Tong, D., Fu, J.S., Hu, X., Lee, P., Meng, X., Chang, H.H. and Liu, Y., 2018. Satellite-Based Daily PM2. 5 Estimates During Fire Seasons in Colorado. Journal of Geophysical Research: Atmospheres, 123(15), pp.8159-8171.

The authors well addressed the other comment in the last review.

---

## Author Response (AR2)

**Review of Xue et al., Satellite-based Estimation of the Impacts of Summertime Wildfires on Particulate Matter Air Quality in United States**

This study uses the GWR method to predict surface PM2.5 concentrations in the US based on satellite AOD and meteorological variables. The statistic method is robust and is well referenced from previous studies, and the prediction results show good agreement with the in-situ measurements. However, the study is still lack of adequate scientific expansion from the results, and the conclusions are similar to the studies on satellite AOD products or ground measurements only, making this study less meaningful.

Before the consideration of publishing, the authors need to further explore the prediction results, make good findings or quantifications that simple AOD and scattering ground measurements cannot show. The authors also need to clean up the minor typos, formats, the potential figure-caption disagreements and misleading journal names in the page head.

We thank the reviewer for the insightful comments, and we have incorporated changes to reflect most of the suggestions provided by the reviewer.

The major difference between the science of AOD and surface PM2.5 is that, AOD is showing the vertical column conditions instead of surface only. Even not considering the aerosol chemistry and secondary formation in clouds, the convection conditions, atmospheric stability or vertical profiles of other meteorological conditions should contribute very much to the difference of AOD and surface PM. Especially for fire plumes, the long-term transport of fire smoke can be at a high altitude, and the vertical pattern of PM will be very different from the no-fire patterns. However, in the GWR model used in this study, only near-surface data are used. Also, noticing the AOD coefficient is much higher that all the other predictors (Table 4). It is doubtful how good the model is, compared to the agreement between AOD and surface PM. Therefore, the authors need to:

1. Show the improvement of the model from using AOD as the only factor, and discuss how the model predict the surface PM out of a column variable.

We added a paragraph describing how the column AOD is related to surface PM2.5:

AOD which represents the total column aerosol mass loading is related to surface $PM_{2.5}$ as a function of aerosol vertical properties and physical properties (Koelemeijer et al., 2006):

$$AOD = PM_{2.5}\, H\, f(RH)\, \frac{3 Q_{ext,dry}}{4\rho\, r_{eff}} = PM_{2.5}\, H\, S \tag{1}$$

Where H is the aerosol layer height, f(RH) is the ratio of ambient and dry extinction coefficients, $Q_{ext,dry}$ is the extinction efficiency under dry conditions, $r_{eff}$ is the particle effective radius, $\rho$ is the aerosol mass density and S is the specific extinction efficiency ($m^2\,g^{-1}$) of the aerosol at ambient conditions. Therefore, AOD usually has a strong positive correlation with $PM_{2.5}$ but the relationship varies depending on other meteorological variables. BLH is used to represent for H by assuming boundary layer is well-mixed under most conditions, and RH related to f(RH) in the equation can adjust the ratio between PM2.5 and AOD through hygroscopic growth of particles. Other meteorological such as surface temperature, wind

speed and surface pressure mainly affect the aerosol mass density through different processes which are introduced in the 2.3 section of the paper.

The improvement of the model (GWR) from using AOD as the only factor has been investigated in other studies (Jiang et al., 2017), which shows improvements of $R^2$ from 0.69 to 0.78 and RMSE from 7.25 to 6.18 by adding 4 meteorological parameters in summer in easter China. For our model, $R^2$ increases from 0.79 to 0.83 and RMSE decreases from 3.8 to 3.4 from the AOD only model. $R^2$ and RMSE has larger improvements for smaller AOD values than AOD larger than 35 $\mu g\ m^{-3}$: $R^2$ increases 0.9 from AOD only model for regions with AOD less than 35 (0.6 to 0.69), while $R^2$ increases 0.05 for areas with AOD larger than 35. RMSE decreases 12% and 7% for AOD less and larger than 35 conditions respectively. Overall, the meteorological factors have larger improvements for low polluted areas.

2. Estimate the model performance only looking at fire region, compare to Figure 3, and discuss the performance and potential bias.

The figures below show the model fitting and validation results for the whole US (left), smoke regions (middle) and NW US (right). The left figure is the original figure 3 in the paper, the middle figure is the results for performing GWR model only for regions with smoke flag larger than 0 and the right figure is for NW US bounded by 35~50°N and 105~130°W. the model performances at fire regions are relative stable and shows similar results compared with the model for whole US region. The $R^2$ have very little variances among the three models, only the RMSE varies due to the higher PM2.5 values close to the fire. This is the benefit of using GWR model to simulate spatially varying factors. The model itself already considered spatial variances, so the model performance at different regions should be the same.

[Figure]

Except for the main concerns, there are some minor suggestions and questions listed below:
1. Line 141-143: Since all the regions in the US are evaluated (in Figure 6), FRP in the other regions should also be verified, to make sure the 2011-2018 difference over the regions other than NW US is not affected by regional fire.

The figures show the PM2.5 difference (left) and FRP difference (right) of the two years in different EPA regions. NW US (EPA region 8~10) has larger FRP in 2018 than in 2011 while all other regions has no fire or has slightly larger FRP in 2011 than in 2018. Therefore, the PM2.5 increase in region 6 due to 2018 fires could be underestimated, and the PM2.5 decrease in region

3~4 is partly due to the fires in 2011. By quantifying the total FRP in different EPA regions, we add some relevant uncertainties in the 4.6 section.

[Figure]

2. Line 269-270: as discussed in 1.2, further discussions e.g. calculation $R_2$ for high PM values may be useful.

As suggested by the reviewer, we have added some discussions for $R^2$ for both high and low PM2.5 values in section 4.2.

3. Line 274-278: Is there any logic about the box selecting? For example, how to decide the size of the box? Can the box be larger? For each type of region, the authors can also show a regional mean with standard deviations of each coefficient. Also, can the box/region selection be more quantified, for example, by classifying using the background PM concentrations or FRP?

We thank the reviewer for pointing this out. The box selecting is based on the smoke transport path and surface PM2.5 concentrations. The red box is selected to include all the fire sources in US (as mentioned in the paper, major fire sources are in 3 states: Washington, Oregon and California), the yellow box is selected based on the estimation map of surface PM2.5 because this region has relative high pollution concentration and no fire sources which means that the smoke is transported from other places. The green box is selected of the same reason as the yellow one but with longer distances from the fire source region which represents for different vertical distribution from the yellow box. the black box is the farthest region from the fires in which the pollution sources should mainly from anthropogenic pollution but not wildfires. The size of red box is the largest so it can cover 3 states and all the fires, while the yellow and green box are the same size large enough to include PM2.5 within certain ranges and the black box size is chosen to include the minimum change regions according to the difference map. Within each of the box, we also select samples according to the estimated PM2.5 and FRP values to calculate the mean and standard deviation of different coefficients.

[Figure]

D-PM2.5 map between 2018 and 2011 August

| Mean coefficients | sample selection | N | AOD | smoke flag | PBL | T2M | RH | U | SP |
|---|---|---|---|---|---|---|---|---|---|
| box1(red) | FRP>1000 | 213 | 91.94 | -0.14 | -2.25 | 0.33 | 0.08 | -2 | -0.06 |
| box2(gold) | PM2.5>30 | 362 | 60.1 | 0.013 | -2.9 | 0.23 | -0.08 | -1.6 | -0.03 |
| box3(green) | PM2.5>17 | 278 | 6.2 | 0.05 | 0.2 | 0.2 | 0.014 | -0.3 | -0.02 |
| box4(black) | 17>PM2.5>10 | 938 | 7.1 | -0.02 | -1.2 | 0.22 | -0.035 | 0.06 | -0.005 |
| whole US region | ~ | 106352 | 28.1 | 0.024 | -0.9 | 0.06 | -0.04 | -0.7 | -0.002 |

| Standard deviation | AOD | smoke flag | PBL | T2M | RH | U | SP |
|---|---|---|---|---|---|---|---|
| box1(red) | 6.5 | 0.04 | 0.5 | 0.3 | 0.15 | 0.45 | 0.03 |
| box2(gold) | 6.1 | 0.02 | 0.11 | 0.07 | 0.02 | 0.07 | 0.004 |
| box3(green) | 0.11 | 0.002 | 0.05 | 0.02 | 0.002 | 0.02 | 0.001 |
| box4(black) | 1.45 | 0.01 | 0.22 | 0.17 | 0.03 | 0.16 | 0.006 |
| whole US region | 29.3 | 0.08 | 1.01 | 0.31 | 0.1 | 0.66 | 0.03 |

4. Line 612-614: The caption of Figure 4 seems not agree with the figure it self. "*PM2.5 values equal or larger than 30 μg m-3 are shown as the same color (red)…*", but the color label is ~-5 to 60 ug/m3.

Corrected.

The "Satellite-based Estimation of the Impacts of Summertime Wildfires on Particulate Matter Air Quality in United States" presents an interesting work, although it requires a major revision before it is suitable for publication in the esteemed journal ACP.

We thank the reviewer for the helpful suggestions and we have revised the paper to address the concerns.

a. The term "Particulate Matter Air Quality" is very confusing here. I have checked the manuscript and I did not see the authors discuss PM10 or other Particulate Matter. Meanwhile, "Air quality" can make people think of "Air quality index". Actually, the authors should specifically emphasize this research is concerning the Impacts of Summertime Wildfires on PM2.5 concentrations. The revised title is more precise and more consistent with the abstract and the manuscript.

Thank you for pointing this out, we have changed the title to PM2.5.

b. Given many relevant papers concerning PM2.5 estimation I read and reviewed, PM2.5-Meteorology interactions are very important factors to be considered and well explained. For the introduction part, the reviewer should briefly state that different meteorological factors influence pm2.5 concentrations through different mechanisms and introduce some relevant studies to present readers a clear background. In the method part, when the authors explain what meteorological factors were included in the research, they should also briefly introduce why or introduce some relevant studies.
Generally, PM2.5-Meteorology relationship is closely related to this research and should definitely be introduced more in details in the manuscript. Authors could refer to some recent references for more information.

As suggested by the reviewer, we have added a paragraph explaining the interactive processes between PM2.5 and meteorological parameters in section 2.3.

c. The format and language of this manuscript should be carefully checked, as there are many unnecessary typos. For instance, this manuscript was not converted to a specific ACP template. Instead, it confusingly shows" for review in remote sensing of environment". And, throughout the manuscript, 2.5 in "PM2.5" is not presented in a subscript format, which should be completely checked and revised. And other language issues should also be addressed through a more careful proof-reading.

We have corrected all the typos pointed out by the reviewer.

---

## Author Response (AR3)

Sundar A. Christopher, Ph.D.
Professor, Department of Atmospheric and Earth Science
Huntsville, AL
sundar@nsstc.uah.edu

June 7, 2021

To
The Editor
ACP

Dear Editor,

We are submitting a revised version of the manuscript MS No.: acp-2020-1152.

Reviewer 2 suggested accepting the manuscript with minor revisions.

Reviewer 1 suggested that we include the material from the response to his suggestions in the manuscript. We have added material to the revised manuscript and added a supplementary section explaining further details.

We believe that the manuscript is now ready for publication.

We look forward to a positive response.

Sundar Christopher.

Sincerely,

*Sundar. A.C.*

Sundar A. Christopher
Professor

**Sundar A. Christopher, Department of Atmospheric and Earth Science, 320 Sparkman Drive, NSSTC, Huntsville, Alabama 35805, Phone: +1-256-961-7872| E-mail: sundar@nsstc.uah.edu**
**nsstc.uah.edu/sundar**

Second review of Xue et al., Satellite-based Estimation of the Impacts of Summertime Wildfires on Particulate Matter Air Quality in United States

The authors did not address the major issue I pointed out in the first response, and the revised manuscript did not add any information supporting a deeper finding or importance of this work. Some of the questions are well answered, which should be good for the scientific significance, but the corresponding updates did not appear in the revised manuscript. The authors need to further consider this issue and do full discussions in the manuscript.

We thank the reviewer for the insightful comments, and we have been able to incorporate changes to reflect most of the suggestions provided by the reviewer. We have added additional material in a supplement section of the revised manuscript.

Before discussing the old issue: although it may not be a reviewer's responsibility, the font size in Section 4.3 is smaller than the previous test, and the acknowledgment did not show credits to the in-situ PM2.5 measurements or the ECMWF products. Keeping these errors and typos will hurt the credibility of the journal.

We the reviewer for pointing this out. The font size is corrected, and the acknowledgment is added for both EPA and ECMWF data.

1. For the response to my major issue 1, the response is not describing how the column AOD is related to PM2.5. Here is the reason:
Here shows the equation (1) the authors added:

$$AOD = PM_{2.5} \, H \, f(RH) \frac{3Q_{ext,dry}}{4\rho \, r_{eff}} = PM_{2.5} \, H \, S$$

This equation represents the AOD in a specific layer. If 'H' is the height of the column, the 'AOD' is the column AOD; if 'H' is the PBLH, then the 'AOD' on the LHS is just an AOD of the PBL. Thus, when in the response the authors claimed that H represents PBLH, the AOD is not the total column from satellite data. This claim did not appear in the manuscript. Therefore, the main issue pointed out in the last review was not solved. AOD does have high correlation with PM2.5 because usually the vertical PM2.5 profile does not significantly change, and the surface source of PM2.5 is mostly mixed in the BL. It is right in most cases, but not in the fire cases. The average fire plume height is 1-2 km, and sometimes the plume can go to 5 km, or even higher. When this study is focusing on wildfires, this issue cannot be ignored, and the free-troposphere transport of smoke will be a robust bias in the model, because the model did not include any parameters with the vertical information. This probably leads to the underestimation of the prediction of the large values. The authors need to well address and discuss this issue in the manuscript.

We thank the reviewer for pointing this out. we add some explanation in section 2.3 and some discussion in 4.7 about the uncertainties. We tried to use the plume height (smoke injection height in MAIAC product MCD19A2) as one input to the GWR model, and it increased the R value from 9.13 to 9.14 but decrease the R for validation from 0.89 to 0.88. For both smoke regions (smoke flag>0) and NW US region, adding the plume height information leads to prediction accuracy decreases. We also compared the plume height product (below left figure) with the CALIPSO vertical profile (below right figure) on August 19th 2018, and the comparison shows that MAIAC

underestimates plume height. therefore, we decide not to include the plume height in the model. It is difficult for passive sensors such as MODIS to calculate plume heights.

[Figure]

The second part of this response discussing the previous studies is good for stating the significance of the model and the study, which also answers another reviewer's question about the correlation with meteorological data, but it seems not in the manuscript. This also helps the discussion in the previous paragraph: when the vertical profile information of PM is missing in the model, the weight of parameters other than AOD should be higher. The authors need to discuss this in the manuscript with Table 4, showing the agreement or disagreement about the AOD weight in this study and previous studies.

We added the discussion in section 4.3. The actual weighting is hard to compare with other studies since the coefficient differs for different cases, so we just use different regions in our own model to illustrate the vertical distribution information can improve the model compared with regions with no vertical information (where aerosol higher than BLH).

2. The figures showing the smoke impact region and NW US addressed my question, but not showing in the manuscript. If the authors prefer a clear main text, I recommend the authors include this in the supplementary materials, because without it, it will be the reader's concern that the low values from regions free from fire smoke may dominates the high R.

We added the contents in the supplementary materials.

3. In addition to the previous review and response, the authors need to compare the method and model performance with previous studies.

We added a section (4.6) to compare our method and results with previous studies.

---

## Author Response (AR4)

Since this is a revised version, previous reviewers have put forward a lot of comments and the authors have made a lot of efforts to modify the paper. Now the study has been largely improved, but I still have some suggestions for the author's reference:

We thank the reviewer for the helpful comments and efforts towards improving our manuscript.

Abstract: The authors should provide quantitative descriptions of the overall accuracy of PM2.5 estimations in the US using the GWR model.
We have added the quantitative description of the overall accuracy of GWR model.

Introduction: Besides the chemical and statistical regression methods, it is suggested that the authors should summarize recent studies on PM2.5 estimation from MODIS AOD products using the popular artificial intelligence methods (Hu et al., 2017; Li et al., 2017; Wei et al., 2019, 2020, 2021).
We have added the machine learning method in the introduction.

Line 140-141: Reference for ERA5 is needed.

The reference for ERA5 is added.

Line 183: Why only select PM2.5 > 2.0 μm/m3? Any reason? There are a lot of PM2.5 values below 2 μm/m3 in the US (Hu et al., 2017).

The reason for discard PM2.5 is that the established lower detection limit for FRM method is 2 μm/m3 for 2011 (3 μm/m3 for 2018) which is explained in section 2.1.

Line 144-166: How about the precipitation? We know that there is usually a lot of rain in the United States, which can clean the air pollution (Wei et al., 2020).

We added some explanation in the end of section 2.3 stating that precipitation could decrease the PM2.5 concentration but since AOD data is usually invalid when raining, we did not use precipitation as a predictor in this study.

Section 4.2: How about the accuracy of PM2.5 estimates in 2011 since you have compared the results with the year 2018 later?
The results for PM2.5 estimation in 2011 is added in the supplement.

The Figures can be improved, e.g., Figure 3, remove R, and add RMSE;
Figure 4: Units for the legend are missing;
Please add the Y-axis title and unit in Figure 5.
We thank the reviewer for pointing this out. We have corrected figure 3 to figure 5.